# Fitness landscape of substrate-adaptive mutations in evolved amino acid-polyamine-organocation transporters

**Foteini Karapanagioti, Úlfur Águst Atlason, Dirk J Slotboom\*, Bert Poolman\*, Sebastian Obermaier\***

Department of Biochemistry, University of Groningen, Groningen, Netherlands

**\*For correspondence:**
d.j.slotboom@rug.nl (DJS);
b.poolman@rug.nl (BP);
s.obermaier@protonmail.com
(SO)

**Competing interest:** The authors declare that no competing interests exist.

**Abstract** The emergence of new protein functions is crucial for the evolution of organisms. This process has been extensively researched for soluble enzymes, but it is largely unexplored for membrane transporters, even though the ability to acquire new nutrients from a changing environment requires evolvability of transport functions. Here, we demonstrate the importance of environmental pressure in obtaining a new activity or altering a promiscuous activity in members of the amino acid-polyamine-organocation (APC)-type yeast amino acid transporters family. We identify APC members that have broader substrate spectra than previously described. Using in vivo experimental evolution, we evolve two of these transporter genes, *AGP1* and *PUT4*, toward new substrate specificities. Single mutations on these transporters are found to be sufficient for expanding the substrate range of the proteins, while retaining the capacity to transport all original substrates. Nonetheless, each adaptive mutation comes with a distinct effect on the fitness for each of the original substrates, illustrating a trade-off between the ancestral and evolved functions. Collectively, our findings reveal how substrate-adaptive mutations in membrane transporters contribute to fitness and provide insights into how organisms can use transporter evolution to explore new ecological niches.

## eLife assessment

This **important** manuscript describes experimental evolution experiments using a novel genetic system in yeast, showing that solute carrier transporters can incorporate additional functionality through the introduction of point mutations to either the ligand binding site or gating helices. These findings provide **convincing** evidence to establish that for Amino Acid transporters of the APC-type family, evolution to recognize new substrates passes through generalist intermediates that can transport most amino acids.

## Introduction

Life continuously creates genetic innovation, e.g., genes coding for proteins with novel functions. Most new genes emerge by modification of existing genes rather than de novo (*Long et al., 2013*). The first who connected the generation of new functions with the accumulation of mutations and duplication of pre-existing genes was *Muller, 1936*. Later on, different theories arose discussing the point at which divergence takes place (*Ohno, 1970*; *Jensen, 1976*; *Lynch and Force, 2000*; *Bergthorsson et al., 2007*). Modern evolutionary models assume an ancestral gene to acquire a novel function while retaining its original function (*Bergthorsson et al., 2007*; *Hughes, 1994*; *Aharoni et al., 2005*; *Des Marais and Rausher, 2008*; *Näsvall et al., 2012*). If a novel function is beneficial for the organism, the gene variant is subjected to positive selection. Consecutive duplication of the

gene allows the establishment of the new function in the population. This model of divergence prior to gene duplication allows for the accumulation and selection of mutations with adaptive potential while the original gene function is maintained, thus creating a 'generalist' gene with multiple functions (*Soskine and Tawfik, 2010*).

When considering the evolution of novel gene functions, the substrate range of 'classical', soluble enzymes is arguably the most archetypal field of study, important for the rapid evolution of antibiotic resistance, degradation of artificial xenobiotics, or use of new nutrients (*Aharoni et al., 2005*; *Copley et al., 2023*). The substrate spectrum of an enzyme is often a key determinant of the fitness of an organism. Generally, enzymes are thought of as being highly specific for a certain substrate or a group of chemically similar substrates. Nonetheless, many enzymes studied in isolation show side activities toward other substrates (*Copley et al., 2023*; *Khersonsky and Tawfik, 2010*; *Andorfer et al., 2017*). These side activities are often obscured in the complex setting of a biological cell, and as such contribute little to the fitness of the organism. However, these side activities are thought of as a major source for genetic innovation: proteins can evolve the ability to use new substrates by improving an already existing, but low, side activity (*Janssen et al., 2005*; *Jørgensen et al., 2017*; *Caligiore et al., 2022*; *Zheng et al., 2020*).

Membrane transporters form a special class of enzymes, as they generally do not break or make covalent bonds. Hence, they experience fewer constraints of the chemistry of their substrates. In principle, any small molecule can be transported through a lipid membrane. Thus, it is not surprising that life has evolved transport systems for virtually every metabolite, ion, and xenobiotic (*Saier et al., 2014*; *Saier et al., 2021*). In many branches of the tree of life, transporter genes underwent processes of repeated duplication and divergence, leading to extended gene families; many genomes have an array of transporter paralogs (*Copley et al., 2023*; *Saier et al., 2021*; *Jack et al., 2000*; *Perland and Fredriksson, 2017*). Typically, each paralog displays a distinct substrate specificity profile, and thus a distinctly evolved function. Some notable examples of gene family expansion are found within the APC (amino acid-polyamine-organocation) superfamily of transporters, which include: the neurotransmitter transporter family in animals (20 paralogs in human, TC 2.A.22), the AAAP family in plants (58 paralogs in rice, TC 2.A.18), the AAT family in Proteobacteria (11 paralogs in *Escherichia coli*, TC 2.A.3.1), and the yeast amino acid transporter (YAT) family in fungi (18 paralogs in budding yeast, TC 2.A.3.10). The APC superfamily is the second-largest group of membrane transport proteins (*Vastermark et al., 2014*).

Most APC superfamily members are transporters for amino acids, their analogs, or related amines. Some transporters are generalists, accepting a wide range of substrates; others are considered specialists, recognizing one or a few related substrates (*Jack et al., 2000*; *Bianchi et al., 2019*). How do such patterns arise? Does the evolution of specialists move through a generalist stage? Or do new specialists evolve through switching from one specific substrate to another? For classical enzymes, many lines of evidence point to the former (*Aharoni et al., 2005*; *Copley et al., 2023*; *Khersonsky and Tawfik, 2010*; *Risso et al., 2013*; *Mascotti et al., 2018*). However, the evolution of new traits in the special case of transporters is understudied, partly because of a lack of appropriate experimental evolution methods (*Bali et al., 2018*).

Here, we propose that APC-type transporters can evolve novel functions through generalist intermediates, similar to classical enzymes. We showcase the presence of so far unknown substrates of five out of seven studied YAT and develop a growth-based selection platform. Utilizing in vivo experimental evolution, under purifying selection for the original function and simultaneous selection for a new function, we evolve the substrate spectrum of two of these transporters. We show that the acquired mutations result either in improving an existing weak activity or in establishing activity for a new substrate. At the same time, each mutation has a distinct impact on the fitness of the organism for each of the transporter's original substrates.

## Results

The budding yeast, *Saccharomyces cerevisiae*, typically encodes 18 members of the YAT family, each with a different substrate profile (*Bianchi et al., 2019*). This array of transporters enables yeast to use 20 different L-amino acids as the sole nitrogen source in minimal media. These include 17 of the 20 proteinogenic amino acids (the exceptions are Cys, His, and Lys) and 3 non-proteinogenic ones (γ-amino butyric acid [GABA], L-ornithine [Orn], and L-citrulline [Cit]). Here, we use an engineered

**Table 1.** Substrate range of transporters used in this study.

Only amino acids showing a significantly higher growth rate than the vector control (ANOVA with Dunnett's test against vector, p<0.05) are shown. Substrates newly identified in this study are underscored. Substrates with an asterisk cannot be used as the sole N-source by *S. cerevisiae*. For *AGP1*, L-citrulline (Cit) was verified as an actual substrate in a separate experiment (*Figure 4—figure supplement 2*) because of a very low growth rate.

| Transporter gene | Uniprot accession number (UniProt Consortium, 2023) | Substrates described in literature (reviewed in Bianchi et al., 2019) | Substrates found in the present study |
|---|---|---|---|
| *AGP1* | P25376 | Ala, Asn, Asp, Cys*, Gln, Glu, Gly, His*, Ile, Leu, Met, Phe, Pro, Ser, Thr, Trp, Tyr, Val, GABA | Ala, Asn, Asp, GABA, Gln, Glu, Ile, Leu, Met, Phe, Pro, Ser, Thr, Trp, Tyr, Val, (Cit) |
| *BAP2* | P38084 | Ala, Cys*, Ile, Leu, Met, Phe, Tyr, Val | Ala, Asn, Asp, Cit, Gln, Glu, Gly, Ile, Leu, Met, Phe, Pro, Ser, Thr, Tyr, Val |
| *CAN1* | P04817 | Arg, His*, Lys*, Orn, Ser | – |
| *HIP1* | P06775 | His* | – |
| *LYP1* | P32487 | Lys*, Met | Ala, Asn, Met, Phe, Ser, Val |
| *MMP1* | Q12372 | *S*-Methylmethionine* | Val |
| *PUT4* | P15380 | Ala, Gly, Pro, GABA | Ala, GABA, Pro, Ser, Val |

yeast strain (Δ10AA) that lacks 10 membrane transporters for amino acids (*Besnard et al., 2016*). This strain is thus severely deficient in the uptake of amino acids, preventing it from using most of them as the nitrogen source. When an amino acid transporter is genetically reintroduced, its substrate profile directly determines which amino acids can support growth. In the following text, the gene names are used in place of the protein names for consistency (e.g. *AGP1* instead of Agp1). The Δ10AA yeast strain expressing different transporter genes was used in all growth and transport assays.

## Substrate spectrum of seven YAT

We investigated the substrate profile of seven YAT, by measuring the growth rates of Δ10AA strains overexpressing each transporter gene from pADHXC3GH, in the presence of 2 mM of each amino acid that served as nitrogen source. We determined growth rates in order to study each transporter's substrate range separately. For five of the seven transporters tested, we found substrates that were not previously reported in literature. These substrates were identified by significantly increased growth rates of Δ10AA expressing the respective transporter in comparison to the strain carrying the empty plasmid (vector control) (*Table 1*). Because of high background growth of the vector control on Arg and Orn (probably due to the activity of the *VBA5* transporter *Shimazu et al., 2012*), these two amino acids were excluded from further analysis. Additionally, some growth of the empty vector control was observed on Gly, Trp, Leu, Met, and Gln, which complicates the interpretation of the data. The observed growth could indicate an unknown substrate specificity of an endogenous transporter. We observed that some of the newly characterized substrates support growth rates in similar ranges as the transporter's substrates already reported in literature, while other substrates support very slow growth rates, which we call weak or promiscuous activities (*Figure 1* and *Figure 1—figure supplement 1*).

For example, *BAP2*, previously reported to transport seven amino acids, was found to support growth on 16 amino acids (specific growth rate μ=0.07–0.31 hr⁻¹), making it a broad-range transporter. The differences on the observed growth rates on each amino acid can be explained by differences in the activity of the overexpressed transporter for these substrates or changes in the expression level of the endogenous transporter. Specifically, the twofold higher growth rate on Leu compared to Pro could be caused by a higher transcription rate of the endogenous transporter in the presence of Leu (*Didion et al., 1996*; *Magasanik and Kaiser, 2002*). Another transporter which showed a broader substrate range than previously reported is *PUT4*. The *PUT4* transporter was found to support fast growth on Ser (μ=0.21 hr⁻¹) and slow growth on Val (μ=0.03 hr⁻¹), in addition to the already reported transport of Ala, Gly, Pro, and GABA. *AGP1*, an already known broad-range transporter, additionally supports very slow growth (μ=0.03 hr⁻¹) on Cit (L-citrulline), which is lower than the growth rate on any

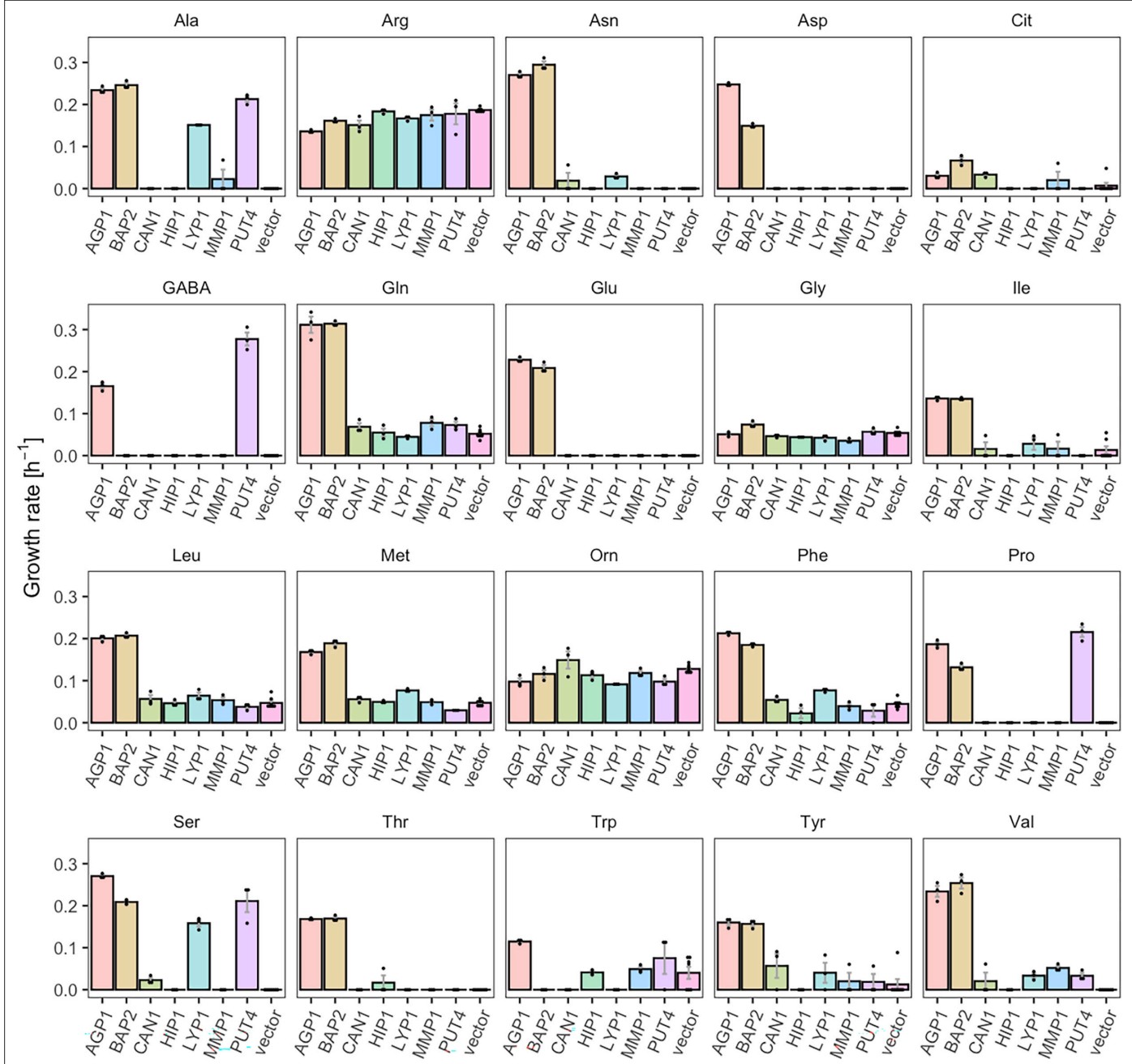

**Figure 1.** Yeast amino acid transporters (YAT) support growth on a range of amino acids. Growth rates of Δ10AA expressing one of seven different wild-type YAT genes (*AGP1*, *BAP2*, *CAN1*, *HIP1*, *LYP1*, *MMP1*, *PUT4*) from pADHXC3GH and the empty vector control on 2 mM of each amino acid. Error bars represent the SEM (n≥3). For the respective growth curves, see *Figure 1—figure supplement 1*. For the respective growth rates, see *Figure 1—source data 1*. To give an indication of the relatedness of YAT proteins from *S. cerevisiae*, a pairwise identity matrix based on Clustal Omega (*Madeira et al., 2022*) alignment of Uniprot (*UniProt Consortium, 2023*) sequences is presented in *Figure 1—figure supplement 2*.

The online version of this article includes the following source data and figure supplement(s) for figure 1:

**Source data 1.** Growth rates of yeast amino acid transporter (YAT) expressing yeast.

**Figure supplement 1.** Yeast amino acid transporters (YAT) support growth on a range of amino acids.

**Figure supplement 2.** Pairwise identities of yeast amino acid transporter (YAT) protein sequences from *S. cerevisiae*.

**Table 2.** Evolved *AGP1*-Cit variants and *PUT4*-Asp and -Glu variants.

| Evolved variant | Nucleotide substitutions (nonsynonymous in bold) | Amino acid substitutions | Abbreviated names of site-directed mutants |
|---|---|---|---|
| *AGP1*-Cit1 | T117C **T1001A** T1191A **A1234G** T1329C C1461G T1899C | I334N (=**N**) I412V (=**V**) | *AGP1*-N |
| *AGP1*-Cit3 | T117C **A308G** T915C **T1001A** C1451G C1461G **T1684C** | D103G I334N (=**N**) A484G (=**G**) F562L | *AGP1*-V *AGP1*-NV *AGP1*-G |
| *AGP1*-Cit11 | T111C **T121C** G1450A A1609G | S41P A484T (=**T**) I537V | *AGP1*-T |
| *PUT4*-Asp1 | **T620C T1474C** | L207S (=**S**) F492L | |
| *PUT4*-Asp2 | **T620C T1415C T1474C** | L207S (=**S**) V472A F492L | |
| *PUT4*-Asp3 | **T191C T620C C734T G956A** | I64T L207S (=**S**) S245F G319D | |
| *PUT4*-Glu1 | G225A **T620C** | L207S (=**S**) | |
| *PUT4*-Glu2 | **T191C T620C G956A** | I64T L207S (=**S**) G319D | |
| *PUT4*-Glu3 | **T191C T620C** | I64T L207S (=**S**) | *PUT4*-S |

of the previously known substrates (µ=0.11–0.31 hr$^{-1}$). As a comparison, growth on ammonium, which does not depend on amino acid transporters, yielded the highest measured growth rate of µ=0.36 hr$^{-1}$ (vector control shown in *Figure 3—figure supplement 1*).

## Experimental evolution of membrane transporter specificity

We tested whether the substrate range of amino acid transporters could be changed by experimental evolution, either by improving promiscuous activities or by evolving toward completely new substrates. For the former, we chose to study *AGP1* under a selective pressure for Cit uptake. For the latter, we studied *PUT4* under selective pressures for uptake of Asp and Glu. For the in vivo evolution, the OrthoRep system was used, which allows for random mutagenesis of a gene while it is actively expressed in yeast (*Ravikumar et al., 2014*; *Ravikumar et al., 2018*). Each of the two transporter genes was encoded on this system and introduced into the transport-deficient Δ10 strain, with additional Δ*ura3* and Δ*his3* deletions needed for selection.

To select for mutants with changed substrate specificity, we employed a dual-selection scheme: The cultures were grown in a minimal medium containing a nitrogen-limiting mixture of natively transported amino acids (1 mM final concentration) to ensure low-level growth of the culture. The mixture of amino acids was composed of equimolar amounts of 17 original substrates for *AGP1* and 2 original substrates for *PUT4*. A concentration of 3 mM of the target substrate was added in the selection for evolved mutants. We expect that in this setup nonfunctional transporters are purged from the evolving population, as these do not support uptake, neither of the original substrates nor of the target amino acid. Transporters with the original substrate range are kept in low numbers, and transporter variants with novel substrates can increase to high numbers as they support growth on the target substrate present at threefold higher concentration than the original substrates.

The cultures of *AGP1* and *PUT4* evolution strains were passaged for multiple generations in the selective media. Colonies that supported growth on the target amino acid were isolated, and the transporter gene was sequenced. The gene was cloned to the pADHXC3GH expression vector and reintroduced into Δ10AA cells. Three *AGP1* variants from Cit selection and three *PUT4* variants each from Asp and Glu selection (*Table 2*) conferred the ability to grow on the respective amino acids (*Figure 2—figure supplement 1* and *Figure 2—figure supplement 2*).

Each evolved *AGP1* variant had multiple nonsynonymous mutations, most of which occurred in the predicted transmembrane part of the protein (*Figure 2* and *Figure 2—figure supplement 3*). In two of the three variants, the I334N (transmembrane helix TM6) mutation was found. The I334 position is located at the permeation pathway of the protein as predicted from AlphaFold (*Jumper et al., 2021*). A mutation in position A484 (TM10) was found in two variants (A484G and A484T). Additionally, mutations were found at the intracellular N-terminus (S41P, D103G) and the predicted transmembrane helices further away from the permeation pathway (I537V in TM11, F562L in TM12).

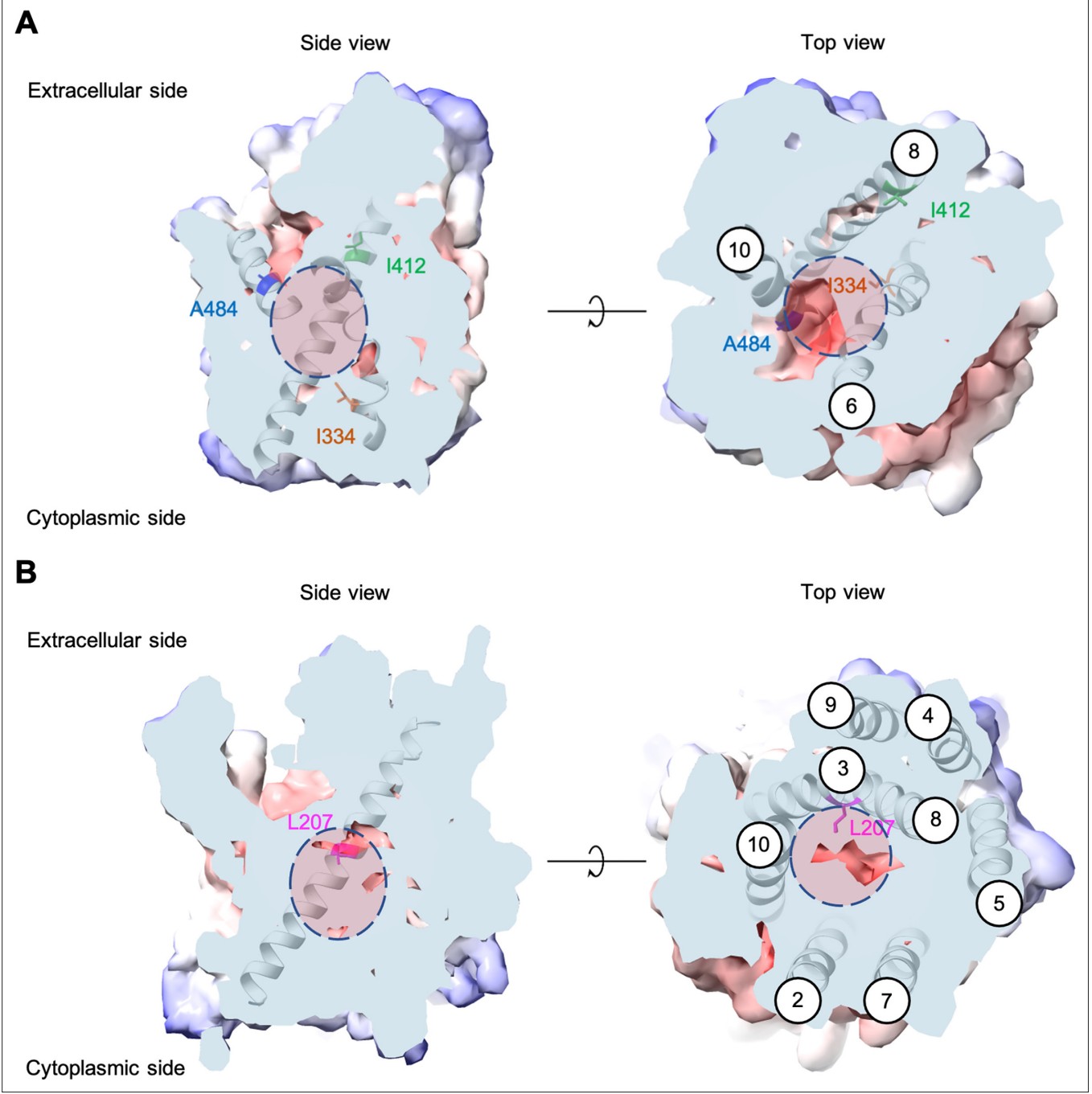

**Figure 2.** Positions of the substituted amino acids investigated in this study. Side and top views of the AlphaFold models of *AGP1* (AF-P25376) (**A**) and *PUT4* (AF-P15380) (**B**) visualized in ChimeraX (1.3.0) (*Pettersen et al., 2021*). The amino acids of interest are highlighted in different colors. The respective TMs are presented in circles. The predicted substrate binding site is represented as dashed circle.

The online version of this article includes the following figure supplement(s) for figure 2:

**Figure supplement 1.** In vivo evolution of *AGP1*.

**Figure supplement 2.** In vivo evolution of *PUT4*.

**Figure supplement 3.** Positions of the substituted amino acids found in the evolved mutants.

**Figure supplement 4.** Position of the L207S on the transporter's binding site.

Most *PUT4* variants had multiple nonsynonymous mutations, again mostly in the predicted transmembrane part of the protein (*Figure 2* and *Figure 2—figure supplement 3*). Interestingly, each variant from both Asp and Glu selections had the same L207S mutation (TM3). Overlay of homologous and structurally similar proteins indicates that this position is part of the substrate-binding site of the transporter (*Figure 2—figure supplement 4*; *Gibrat et al., 1996*). A change from a large hydrophobic residue to a small hydrophilic residue is thus expected to considerably change the properties of the substrate-binding site. Furthermore, in two of the three variants from Asp selection, the F492L mutation (TM10) was found. Mutations were also found at the intracellular N-terminus (I64T), between TM4 and TM5 (S245F), in TM6 (G319D), and in TM9 (V472A).

## Single mutations are sufficient for altered substrate specificity

We wondered whether the altered substrate specificities could be conferred by single mutations. We made site-directed mutations in the wild-type transporter genes and tested the variants for uptake of Cit (*AGP1*) and Asp/Glu (*PUT4*). Generally, only mutations in the transmembrane domains were considered. The shorthand names for each site-directed mutant can be found in *Table 2*.

For the *AGP1* gene, mutations in positions that occurred multiple times were investigated (i.e. I334N, A484G, and A484T; respective abbreviations *AGP1*-N, *AGP1*-G, and *AGP1*-T). Additionally, the I412V mutation was included in the study (shorthand *AGP1*-V), along with a double mutant combining I334N and I412V (abbreviation *AGP1*-NV, recreating the genotype observed in *AGP1*-Cit1). To compare the growth of the mutants on different media, the growth rate of Δ10AA expressing each variant from the plasmid pADHXC3GH was measured. The surface expression of the transporters was estimated by analyzing the fluorescence at the cell envelope (*Figure 3—figure supplement 1A*) and was similar for the wild-type and the evolved proteins (*Figure 3—figure supplement 1B*). Furthermore, there was no statistically significant difference between the growth rate of the strains under conditions where the nitrogen source was not dependent on amino acid uptake (*Figure 3—figure supplement 1C*). Strikingly, when grown in media with Cit as the sole nitrogen source, each of the variants showed a large increase in growth rate relative to the wild-type (*Figure 3A*). Specifically, the growth rate increased 1.9-fold for *AGP1*-N, 2.4-fold for *AGP1*-V, 3.2-fold for *AGP1*-G, and 2.4-fold for *AGP1*-T. The double mutant *AGP1*-NV showed a growth rate increase that is higher than each of its single mutants (2.8-fold). In separate uptake assays with radiolabeled Cit, we found a similar general trend of higher uptake rate of Cit by the mutants (*Figure 3B* and *Figure 3—figure supplement 3A*). We thus conclude that each of the tested single mutations is adaptive for growth on Cit as the nitrogen source.

Regarding the *PUT4* gene, we focused on the L207S single mutation since it occurred in all evolved variants (mutant abbreviation *PUT4*-S). The growth rates of Δ10AA expressing either the wild-type or the *PUT4*-S variant from pADHXC3GH were measured. The growth rate of the strains under conditions where the nitrogen source was not dependent on amino acid uptake was not significantly different (*Figure 3—figure supplement 2C*). While the wild-type transporter conferred no growth in the presence of Asp and Glu, the *PUT4*-S variant allowed growth on both of these substrates, showing that the single mutation is sufficient to change the specificity range of the transporter (*Figure 3C*). To confirm that the observed growth is due to transport of a novel substrate by *PUT4*-S, we conducted a whole-cell uptake assay with radiolabeled Glu. Indeed, Glu was only taken up by the mutant, whereas the wild-type *PUT4* behaved like the vector control (*Figure 3D* and *Figure 3—figure supplement 3B*). Even though the surface expression of *PUT4*-S was higher than that of the wild-type (*Figure 3—figure supplement 2B*), this difference is unlikely to reason for the observed gain of function.

## Single mutations have distinct impacts on the fitness for the original substrates

Above, we established that amino acid transporters can be evolved toward new specificity or increased activity of a substrate by just one mutation. Next, we wanted to know how these adaptive mutations affect the original substrate range and specificity of the transporters. Are the original substrates still transported by the variants? And if so, how does the transport compare to the wild-type? For that, we compared the fitness effects of the *AGP1* and *PUT4* variants for their respective substrates. The relative fitness of each mutant in the studied media was calculated from the growth rate of yeast expressing that mutant, normalized to the growth rate of yeast expressing the wild-type transporter.

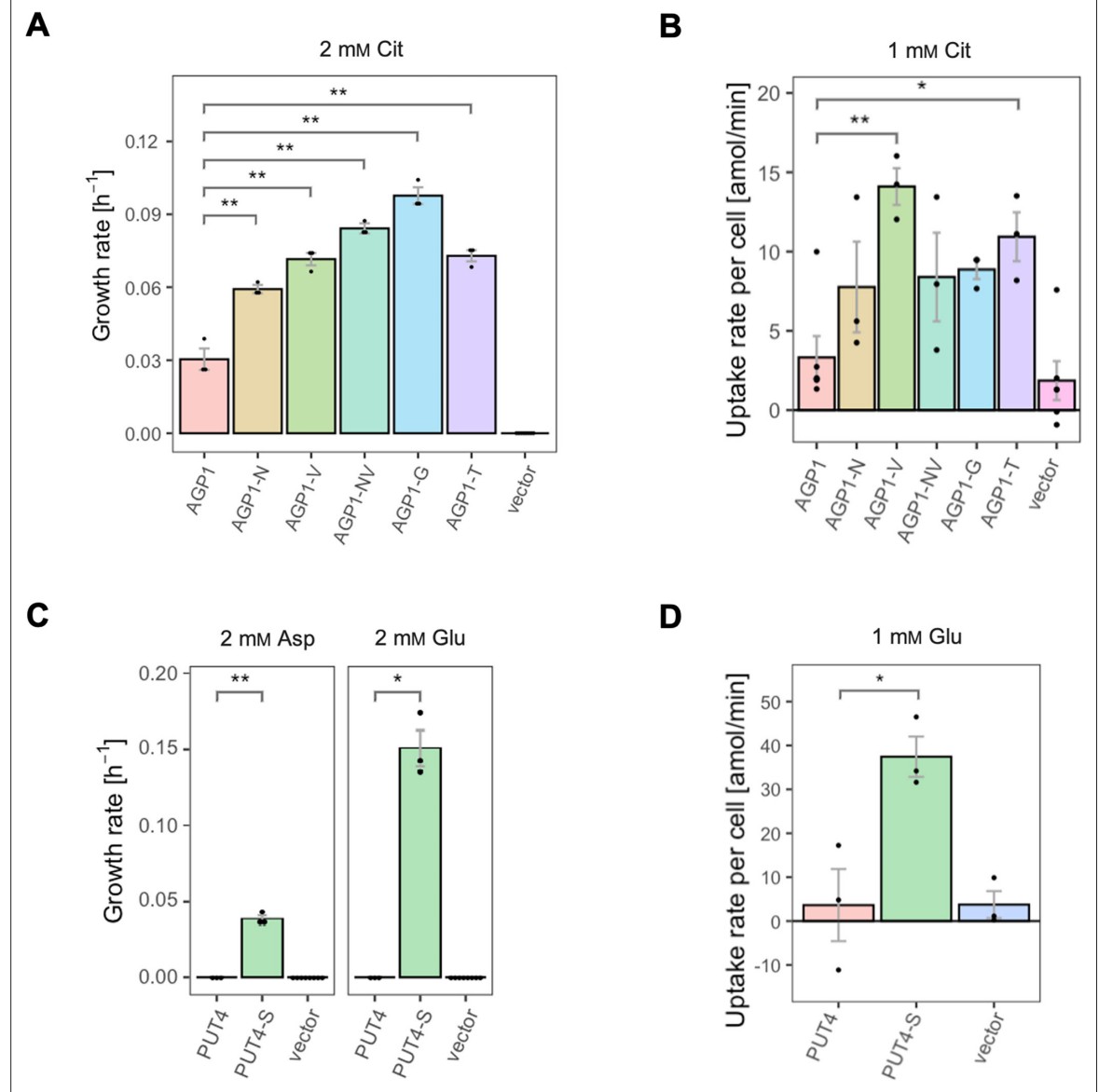

**Figure 3.** The evolved variants support growth and uptake of the respective amino acids. (**A**) Growth rate of the *AGP1* variants and the vector control on 2 mM L-citrulline (Cit). Error bars represent the SEM (n≥3). (**B**) Uptake rate of 1 mM $^{14}$C-Cit by whole cells expressing different *AGP1* variants or none (vector control). Error bars represent the SEM (n≥3). Asterisks in (**A**) and (**B**) indicate the degree of significant difference (one-way ANOVA with a Dunett's test; **p<0.01, *p<0.05) between the *AGP1* variants and wild-type. In case of no significant difference (p>0.05) no asterisks are shown. (**C**) Growth rate of the *PUT4* variants and the vector on 2 mM Asp and Glu. Error bars represent the SEM (n≥3). (**D**) Uptake rate of 1 mM $^{14}$C-Glu by whole cells expressing different *PUT4* variants or none (vector control). Error bars represent the SEM (n≥3). Asterisks in (**C**) and (**D**) indicate the degree of significant difference in pairwise comparisons between the transporter-expressing variants (Student's t-test; **p<0.01, *p<0.05).

The online version of this article includes the following figure supplement(s) for figure 3:

**Figure supplement 1.** Effects of evolved *AGP1* mutations on the surface expression and growth on non-amino acid nitrogen source.

**Figure supplement 2.** Effects of evolved *PUT4*-S mutation on the surface expression and growth on non-amino acid nitrogen source.

**Figure supplement 3.** The evolved variants support uptake of the respective amino acids.

Additionally, we measured the uptake rates of a set of original substrates in whole cells to investigate if the growth fitness effects were due to changed transport activity.

For the *AGP1* variants, we measured growth rates for 17 substrates (*Figure 4* and *Figure 4—figure supplement 1*) and found that none of the single and double mutants had lost the capacity to grow

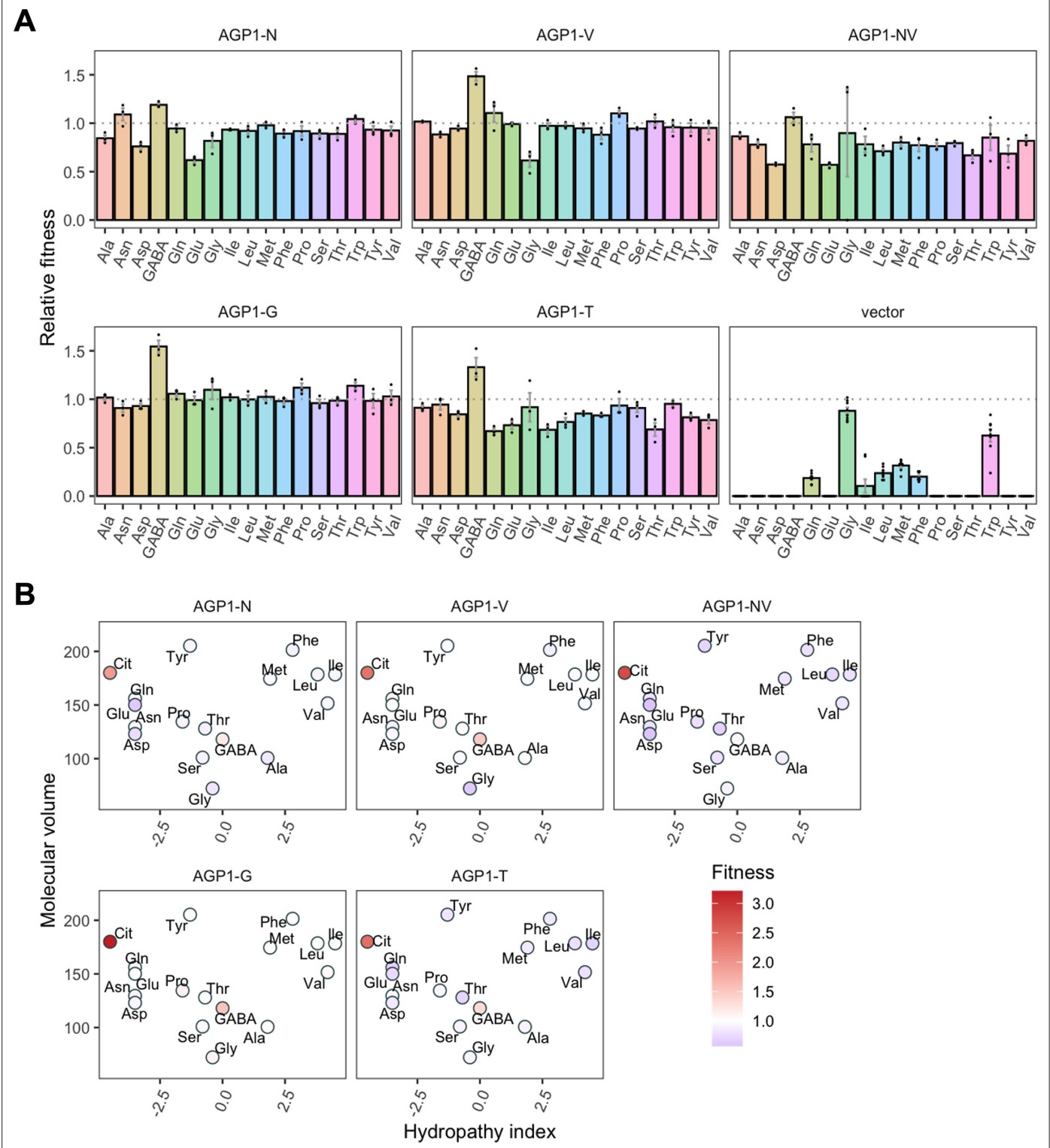

**Figure 4.** Relative fitness of evolved *AGP1* variants. (**A**) Relative fitness of *AGP1* variants and the control strain for the growth on each of 17 amino acids as the sole nitrogen source. The relative fitness was calculated separately for each amino acid by dividing the growth rate of the mutant by the mean growth rate of the wild-type *AGP1*. Error bars represent the SEM (n≥3). For the respective growth curves, see *Figure 4—figure supplement 1*. (**B**) Mean relative fitness of *AGP1* variants per substrate as a function of the amino acid hydropathy index (x axis) and molecular volume (y axis). Color corresponds to relative fitness. The plot is based on the growth rate measurements of panel (**A**).

The online version of this article includes the following figure supplement(s) for figure 4:

*Figure 4 continued on next page*

*Figure 4 continued*

**Figure supplement 1.** The evolved variants affect the strain's growth on different amino acids.

**Figure supplement 2.** Effects of evolved *AGP1* mutations on the growth rate on original substrates.

on any of the original substrates. As a general trend, the *AGP1* variants had a lower fitness than the wild-type, although some had a higher fitness for a given substrate.

The *AGP1* variants had different growth rates on the original substrates (*Table 3* and *Figure 4B*). The amino acid substitutions I334N and I412V had a different effect on the relative fitness depending on whether they were analyzed individually or together in the double mutant. Specifically, the *AGP1*-V variant showed significantly increased relative fitness on GABA as nitrogen source, while showing no significant differences in any of the other substrates. The *AGP1*-N variant showed decreased relative fitness on the negatively charged amino acids Asp and Glu, as well as on Ala. The double mutant *AGP1*-NV showed decreased relative fitness on the majority of the tested amino acids (a total of 12 out of 17), including the ones in which the single mutant *AGP1*-N also exhibited decreased relative fitness. Interestingly, the *AGP1*-NV did not show significantly increased relative fitness in GABA, in contrast to *AGP1*-V. Despite the higher relative fitness of the double mutant on Cit (2.8) compared to that of the single mutants, the variant's low relative fitness on the rest of the amino acids has a greater cost on the overall fitness of that strain (0.78). Also, the observed mistargeting of the transporters to internal membranes in the cases of *AGP1-N* and *AGP1-NV* could indicate misfolding of a fraction of the proteins, which could impact the relative fitness (*Figure 5A*).

The fitness effects were drastically different between *AGP1*-G and -T, both having A484 substituted. The *AGP1*-G variant showed significantly increased relative fitness on GABA and a trend of overall increased relative fitness on small aliphatic amino acids of intermediate hydrophilicity (Pro, Gly). Simultaneously, its relative fitness on Cit was higher than any of the other mutants. *AGP1*-T showed mainly negative relative fitness effects, with decreased relative fitness for charged amino acids and the majority of non-polar amino acids (a total of 9 out of 17).

The idea that the observed fitness effects are due to changed transport activity and/or affinity constant for the substrate was investigated by measuring the uptake rates of radiolabeled Phe and Glu. The uptake was measured at two different amino acid concentrations, namely 0.1 mM and 2 mM, which are respectively lower and higher than the $K_m$ of the wild-type protein for amino acids ($K_m$ = 0.2–0.79 mM [reviewed in *Bianchi et al., 2019*]; *Figure 5* and *Figure 5—figure supplement 1*). For Glu it was found that the uptake rates at both concentrations (*Figure 5C and D*) correlate well with the relative fitness. An almost identical pattern was observed when the uptake and the growth rates of the variants were compared at 2 mM Glu (*Figure 5B and D*). The fact that *AGP1*-N, *AGP1*-NV, and *AGP1*-T showed a pattern of both reduced growth rate and reduced uptake rate compared to the wild-type indicates that the relative fitness indeed is linked to the activity of the transporter (*Figure 5—figure supplement 2B*). Equivalently, in the case of Phe as nitrogen source, the relative fitness and uptake rate of the *AGP1* variants followed similar trends. At 2 mM Phe, the effects on uptake were almost congruent to the growth rates (*Figure 5E and G*, *Figure 5—figure supplement*

**Table 3.** Substrates of *AGP1* strains with statistically significant relative fitness differences.
One-way ANOVA with a Dunnett's test with *AGP1* wild-type as the control group was used with a cutoff of $p<0.05$. Overall relative fitness estimates are calculated from the mean of growth rates on 17 original substrates relative to the wild-type.

| *AGP1* variant | Substrates with significantly increased relative fitness | Substrates with significantly decreased relative fitness | Overall relative fitness (mean of 17 original substrates) | Relative fitness on Cit |
|---|---|---|---|---|
| *AGP1*-N | | Ala, Asp, Glu | 0.92 | 1.9 |
| *AGP1*-V | GABA | | 0.98 | 2.4 |
| *AGP1*-NV | | Ala, Asn, Asp, Glu, Ile, Leu, Met, Phe, Pro, Ser, Thr, Tyr | 0.78 | 2.8 |
| *AGP1*-G | GABA | | 1.0 | 3.2 |
| *AGP1*-T | | Asp, Gln, Glu, Ile, Leu, Met, Phe, Thr, Val | 0.86 | 2.4 |

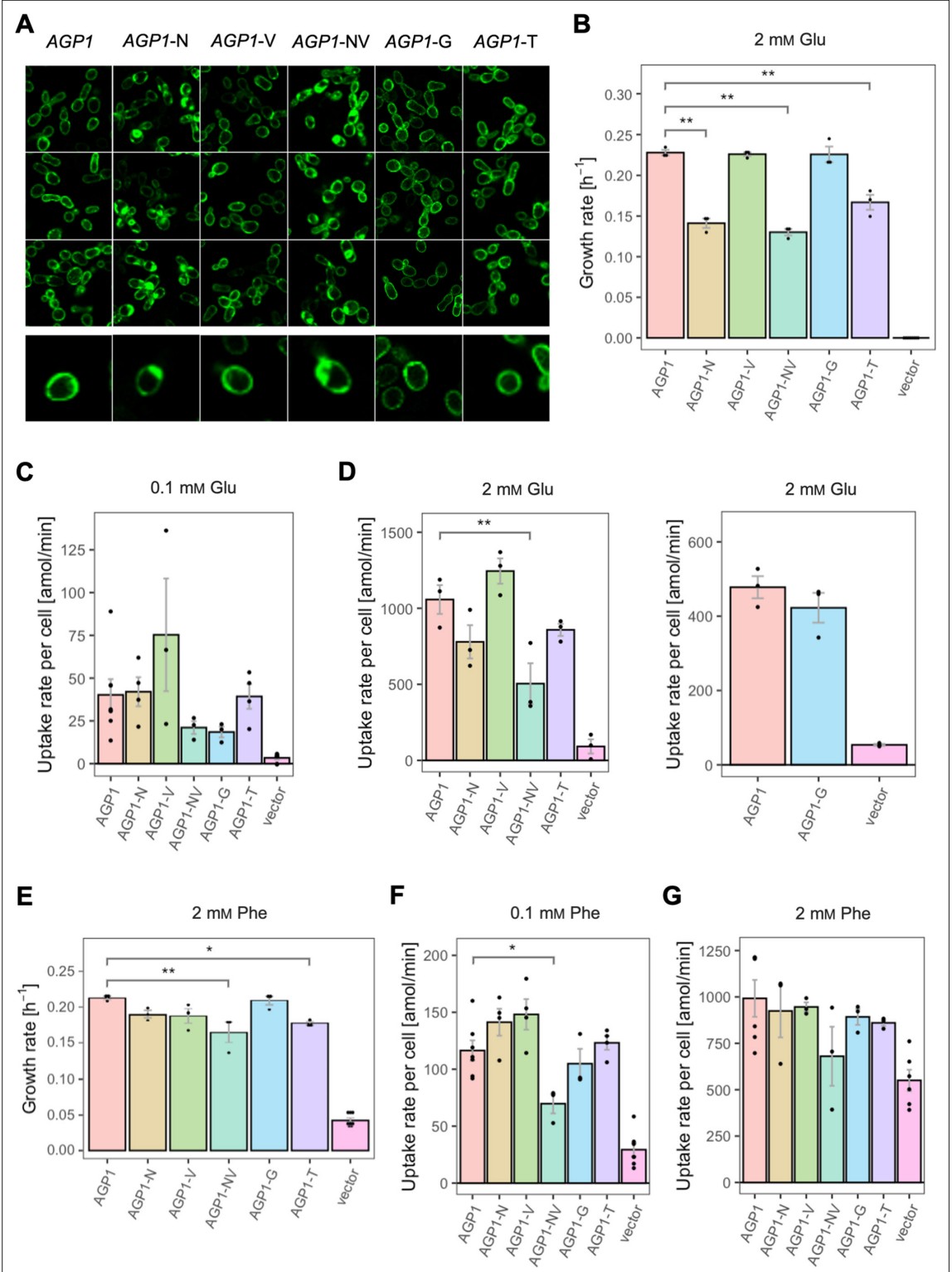

**Figure 5.** Effects of evolved *AGP1* mutations on the growth and uptake of original substrates. (**A**) Localization of the *AGP1* variants in whole cells, based on replicate samples (different colonies) and analyzed by fluorescence microscopy, including a zoom-in of a representative cell. (**B**) Growth rate of the *AGP1* variants and the vector on 2 mM Glu. Error bars represent the SEM (n≥3). (**C–D**) Uptake rate of 0.1 (**C**) and 2 mM (**D**) [14]C-Glu in whole cells expressing different *AGP1* variants or none (vector control). In the case of *AGP1*-G, the assay was performed with 2 mM [3]H-Glu and is presented as independent experiment. Error bars represent the SEM (n≥3). (**E**) Growth rate of the *AGP1* variants and the vector control on 2 mM Phe. Error bars represent the SEM (n≥3). (**F–G**) Uptake rate of 0.1 (**F**) and 2 mM (**G**) [14]C-Phe in whole cells expressing different *AGP1* variants or none (vector control).

*Figure 5 continued on next page*

*Figure 5 continued*

Error bars represent the SEM (n≥3). Asterisks in (**B–G**) indicate the degree of significant difference (one-way ANOVA with a Dunett's test; \*\*p<0.01, \*p<0.05) between the *AGP1* variants and wild-type. In case of no significant difference (p>0.05), no asterisks are shown. For the respective uptake curves, see *Figure 5—figure supplement 1*.

The online version of this article includes the following figure supplement(s) for figure 5:

**Figure supplement 1.** The evolved *AGP1* variants support uptake of original substrates.

**Figure supplement 2.** Correlation between the fitness and uptake of original substrates for the evolved *AGP1* variants.

*2A*). The reduced uptake and growth rate of *AGP1*-NV compared to the wild-type indicates that the lower fitness of the *AGP1*-NV on Phe possibly stems from its reduced transport rate, but an effect of the internalization of the transporter cannot be excluded (*Figure 5A*).

The growth rates of Δ10AA expressing *PUT4* and *PUT4*-S showed that the mutation considerably affects the relative fitness of the strain in three of the original substrates (*Figure 6B*). Specifically, *PUT4*-S was associated with a significant fitness loss during growth on 2 mM Ala and GABA, and a fitness gain in Val. To investigate the phenotype further, we followed the uptake of radiolabeled Ala and GABA at a concentration of 10 µM, well below the reported $K_m$ (reviewed in *Bianchi et al., 2019*; *Figure 6* and *Figure 6—figure supplement 1*). As with *AGP1* variants, the relative fitness and uptake rate trends coincided, indicating that the slower amino acid uptake may cause the slower growth of *PUT4*-S on both Ala and GABA (*Figure 6C and D*). Additionally, we investigated the uptake of Gly, another known substrate of *PUT4*. Due to high background growth of the Δ10AA strain on Gly, there was no significant difference in the growth rates between *PUT4*-expressing and vector control cultures. However, the uptake assays showed a clear transport activity of the wild-type *PUT4* strain, which decreased dramatically in the case of *PUT4*-S (*Figure 6E*). There is thus a clear cost of the mutation on the uptake of Gly.

The wild-type transporter supported very slow growth on Val (µ=0.03 hr⁻¹) and we therefore consider its transport a promiscuous activity. This coincides with the larger size and hydrophobicity of the Val molecule relative to Ala, GABA, Gly, Pro, and Ser. For Val, the fitness of *PUT4*-S was dramatically increased compared to wild-type (2.4-fold), which could be explained with the predicted size increase of the substrate-binding site of the transporter (see above). Thus, the gain-of-function mutation L207S affected the transport of at least four original substrates, three of which negatively, and one positively.

## Single mutations can significantly broaden the substrate range of amino acid transporters

All variants with increased fitness for one substrate can still transport the wild-type's original substrates. For *AGP1*, each of the four tested single mutants as well as the double mutant retained the ability to confer growth on all amino acids that we tested.

Similarly, for *PUT4*, the L207S mutation added transport of Glu and Asp, while still retaining the ability to transport the original substrates Ala, GABA, Gly, Pro, Ser, and Val. Since we established that one single evolutionary mutation extended the strain's growth by two additional amino acids, we wondered whether this mutation also impacted the overall substrate range of the transporter. Therefore, growth assays for the *PUT4*-S strain were conducted with 18 amino acids (*Figure 7A*). Remarkably, the transporter variant conferred growth on 15 of the studied substrates. Specifically, the evolved variant carrying the L207S mutation transports Asp and Glu (the amino acids used for the initial selection), as well as Asn, Cit, Gln, Thr, Ile, Leu, and Met in addition to the six substrates of the wild-type transporter *PUT4*. Regarding the growth on Gln, we note that although the calculated growth rates of *PUT4*-S and *PUT4* were not significantly different, the shape of the growth curve implies that Gln is indeed taken up (*Figure 4—figure supplement 1*). In the case of growth on Phe and Trp, the results were not conclusive due to the high background growth of the vector control strain.

Upon grouping the amino acids according to their size and hydropathy index (*Kyte and Doolittle, 1982*; *Zamyatnin, 1972*), a clear trend emerged for the novel substrates of *PUT4*-S. In addition to the small substrates of low and intermediate hydrophilicity, the evolved variant facilitates the uptake of larger hydrophobic amino acids (Ile, Leu, Met) as well as hydrophilic amino acids (Thr, Asp, Asn, Gln, Glu, Cit) (*Figure 7B*, colored areas 2 and 1, respectively). Thus, we showcase that a single adaptive

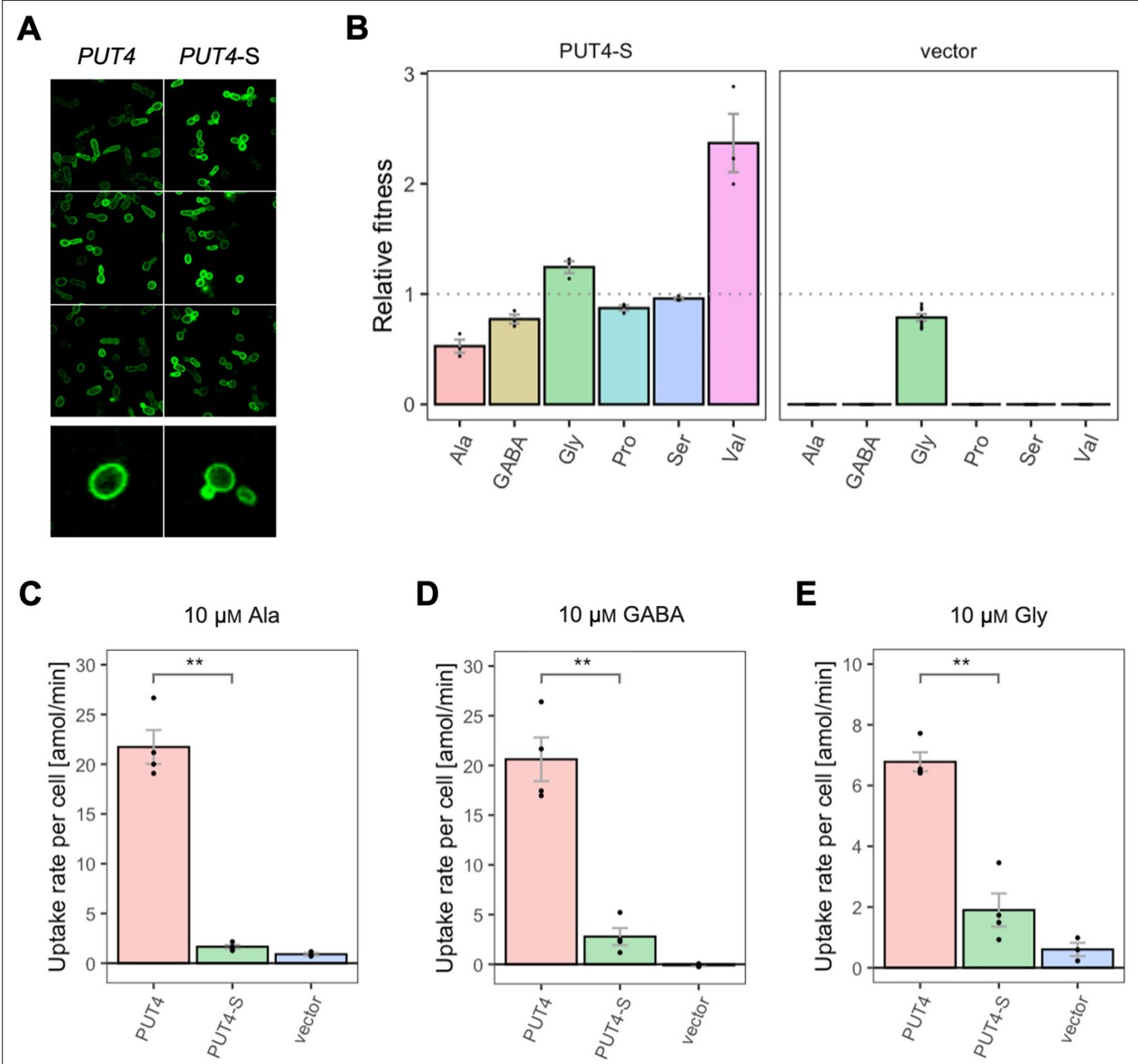

**Figure 6.** Effects of evolved *PUT4*-S mutation on the growth and uptake of original substrates. (**A**) Localization of the *PUT4* variants in whole cells, based on replicate samples (different colonies) and analyzed by fluorescence microscopy, including a zoom-in of a representative cell. (**B**) Relative fitness of *PUT4*-S and the vector for the growth on six amino acids as the sole nitrogen source. The relative fitness was calculated separately for each amino acid by dividing the growth rate of the mutant by the mean growth rate of the wild-type *PUT4*. Error bars represent the SEM ($n \geq 3$). For the respective growth curves, see *Figure 4—figure supplement 1*. (**C–E**) Uptake rate of 10 µM [14]C-Ala (**C**), γ-amino butyric acid (GABA) (**D**), or Gly (**E**) in whole cells expressing *PUT4* variants or none (vector). Error bars represent the SEM ($n \geq 3$). Asterisks in (**C–E**) indicate the degree of significant difference in pairwise comparisons between the transporter-expressing variants (Student's t-test; **$p<0.01$, *$p<0.05$). For the respective uptake curves, see *Figure 6—figure supplement 1*.

The online version of this article includes the following figure supplement(s) for figure 6:

**Figure supplement 1.** The evolved *PUT4* variant supports uptake of original substrates.

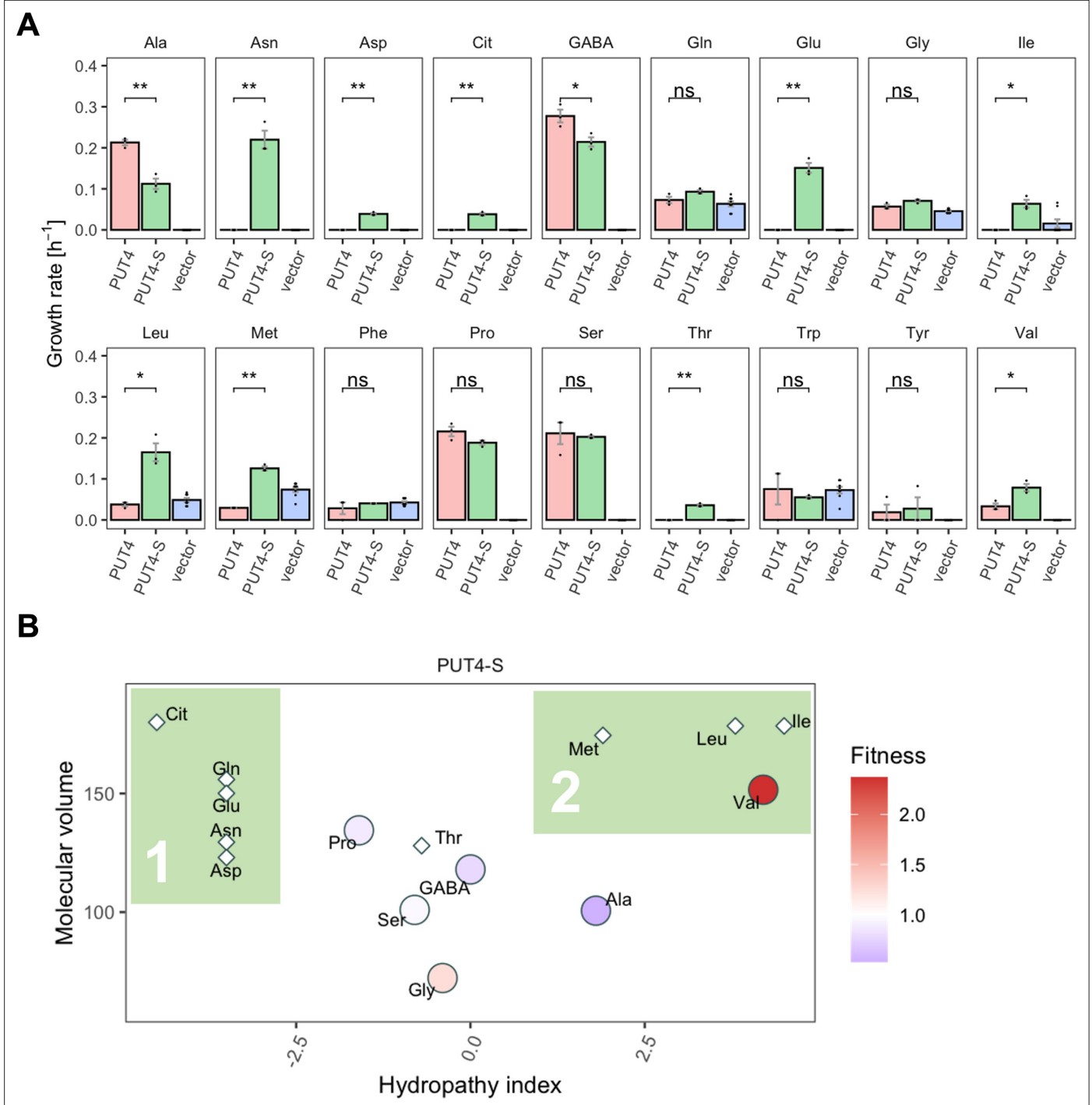

**Figure 7.** The evolved *PUT4*-S mutation broadens the substrate range of the transporter. (**A**) Growth rate measurements of *PUT4* variants and the vector control on 2 mM of 18 different amino acids as the sole nitrogen source. Error bars represent the SEM (n≥3). Asterisks indicate the degree of significant difference in pairwise comparisons (Student's t-test; **p<0.01, *p<0.05). For the respective growth curves, see *Figure 4—figure supplement 1*. (**B**) Mean relative fitness of *PUT4*-S per substrate as a function of the amino acid hydropathy index (x axis) and molecular volume (y axis). Diamond shapes indicate novel substrates not found in the wild-type *PUT4*. Color corresponds to relative fitness. The plot is based on the same growth rate measurements as (**A**).

mutation is sufficient to evolve a narrow-range transporter (*PUT4*: 6 substrates) to a broad-range transporter (*PUT4*-S: 15 substrates).

## Discussion

Different theoretical models have been developed to explain the origin of new functions in proteins. Many favor the idea that new functions are acquired prior to gene duplication while the original functions are maintained (*Soskine and Tawfik, 2010*). While these models have been used to describe the evolution of functions in soluble enzymes, we showcase that the theory also applies to membrane transporters. To simulate natural transporter evolution, we leveraged in vivo evolution of two YAT, *AGP1* and *PUT4*, with selective pressure for transport of novel substrates.

We show that the substrate spectrum of five wild-type YAT is substantially broader than previously described. The use of a growth-based screening process enabled us to identify so far unknown substrates (up to nine in the case of *BAP2*). The transporter genes were expressed from a multi-copy plasmid (2μ origin of replication), where the number of plasmid copies that each daughter cell receives is stochastic (*Chan et al., 2013*). Cells with different copy numbers could potentially have an advantage in our growth rate-based measurements of transporter activity. Thus, the mean copy number of the culture is expected to change during the 3-day cultivation period. The growth rate is therefore a measure of the fitness of the strain when the transporter gene copy number is optimized, which is probably the reason why the growth-based assay is so sensitive for promiscuous transporter activities that have not been described before.

It is apparent that the relative fitness of each strain expressing a transporter depends on the amino acid provided. For the strain expressing *AGP1*, Cit supports very low growth rates indicative of a promiscuous uptake activity by *AGP1* that has not been reported previously. We show that upon the presence of selective pressure for the uptake of Cit, this weak transport activity of *AGP1* is increased by adaptive mutations. Each of the five studied mutants resulted in a gain of relative fitness for growth on Cit. The differential Cit uptake rates of the single and double mutants showcase how in vivo experimentally generated single mutations can alter the substrate spectrum of the YAT without the loss of ancestral functions. Our observations are in line with studies of substrate promiscuity being the result of adaptive mutations in different membrane transporters (*Regenberg and Kielland-Brandt, 2001*; *Tutulan-Cunita et al., 2005*; *Adler and Bibi, 2005*; *Gros and Schuldiner, 2010*; *Ghaddar et al., 2014*; *Young et al., 2014*; *Meier et al., 2023*). The altered specificity for a substrate, which is mirrored in our case in the altered fitness, could be a result of a different affinity constant for the amino acid and/or maximal transport rate.

We also established that through applying evolutionary pressure upon the *PUT4* membrane transporter, single mutations can emerge in the population which provide a gain of function. The newly obtained function is essential for the survival of the organism in the new environment. Interestingly, the same mutation (L207S) in the *PUT4* gene was found when evolutionary pressure was applied for the growth on media supplemented with either Asp or Glu as the sole nitrogen source. The L207 residue is predicted to be part of the substrate-binding site of the transporter. Structurally equivalent sites on other APC superfamily transporters have been shown to affect the function of the studied proteins (*Edwards et al., 2018*; *Ghaddar et al., 2014*). Specifically, the non-conserved V104 in *Aquifex aeolicus* LeuT shapes the binding pocket and influences the substrate specificity (*Singh et al., 2008*; *Yamashita et al., 2005*), while T180 in *CAN1* is expected to contribute to the transporter's interaction with its arginine substrate (*Ghaddar et al., 2014*; *Gournas et al., 2015*). In the case of *PUT4*-S, the replacement of the large hydrophobic leucine residue with a small hydrophilic serine residue is likely to increase both the size and the hydrophilicity of the binding pocket. This could be the reason that the evolved transporter now facilitates the uptake of hydrophilic as well as large hydrophobic amino acids.

The evolved mutations in *AGP1* are also positioned in the transmembrane helices that create the permeation pathway of LeuT-fold transporters (*Kazmier et al., 2017*). I334 in TM6 of *AGP1* is conserved in some members of the YAT family (*DIP5*, *CAN1*, *LYP1*, *ALP1*, *TAT3*). I412 in TM8 is part of the conserved sequence [IV]-L-[ITAVL] and thus in *AGP1*-V and *AGP1*-NV, Ile is changed to the only other common residue, Val, which is found in *DIP5*, *LYP1*, and *ALP1* (*Gournas et al., 2015*). Reportedly, single site substitutions in conserved regions of *CAN1* and *GNP1* have been shown to broaden the substrate range of these YAT and accommodate for the uptake of Cit (*Regenberg and*

*Kielland-Brandt, 2001*). Substitutions in non-conserved regions can also have an effect on a transporter's substrate spectrum. The non-conserved T456 in *CAN1* is important for the substrate specificity of the protein. Single site substitution of T456 can broaden the substrate spectrum of *CAN1* by one amino acid, while S176N/T456S abolishes the uptake of arginine, a high affinity substrate of *CAN1*, and supports high affinity for the uptake of lysine similar to that of *LYP1* (*Ghaddar et al., 2014*). Interestingly, T456 lies two residues away from N454 in *CAN1*, whose equivalent position is A484 in *AGP1* (*Gournas et al., 2015*).

In both *AGP1* and *PUT4*, the evolved variants retain their original substrate spectrum albeit with altered specificities. Our results showcase that specific adaptive mutations for a novel substrate often decrease the fitness for the original substrates, creating a situation of trade-off between new and old function. However, in some cases, the fitness for single original substrates actually increases (e.g. *AGP1*-G and *AGP1*-V for GABA, or *PUT4*-S for Val). Depending on the environment, a mutation can thus be adaptive for multiple substrates at the same time, including ones that were not selected for. For evolution on Cit, the mutant with the highest fitness gain (*AGP1*-G) showed no loss of relative fitness for the original substrates. Thus, this mutation could potentially occur in the population even without the presence of Cit as selective pressure or even outcompete others in a natural setting where the selective pressure is applied. On the contrary, the combination of two adaptive mutations (*AGP1*-NV) had a higher fitness cost for original substrates than the separate mutations, but also a higher fitness gain for the novel substrate, which can be viewed as the first step toward specialization. Thus, in a natural setting, such a variant would only be viable if the novel substrate gives a very high selective advantage as compared to the original ones. Alternatively, gene duplication could lead to two distinct transporter genes, one specializing in the novel substrate, while the other (re-)specializes for the original substrates (*Soskine and Tawfik, 2010*). To generalize our findings to natural evolution, it is important to keep in mind that our selection process was intentionally designed to emulate a *newly encountered substrate*, for which no suitable transporter is yet available. This led to evolution toward substrate range expansion and produced broad-spectrum transporter variants. In wild-type organisms, broad specificities are not always favorable. Each organism typically possesses multiple nutrient transporters with partial redundancy in their substrates, making it possible to finely tune-up and down-regulate specific transport activities according to the nutrient availability (*Bianchi et al., 2019*). Therefore, our experimentally derived transporter variants would probably not fare well in natural settings, as they come from an organism that already has evolved a full suite of amino acid transporters. They do, however, give information about evolvability of transporters when faced with a new solute, whose transport leads to increased fitness of the organism. An example of a transporter with a very special substrate can be found in the cystine transporter *CYN1*, which is distantly related to *LYP1*. Only a fraction of all yeast species has orthologs of this transporter and the respective cystine uptake activity (*Yadav and Bachhawat, 2011*). Seemingly, not all yeasts profit from the ability to assimilate cystine from the medium, and therefore do not possess a suitable transporter. At one point in time, however, an ancestor of *LYP1* evolved toward transport of cystine and ultimately turned into the highly selective *CYN1* found today. These processes take place over long periods of time and are hard to observe in the field. Our experiments emulate the first steps in such an evolutionary scenario and make them observable within laboratory time scale.

Taken together, our findings show that transporters can have promiscuous weak activities, which are often hidden in the natural context of the cell. Furthermore, transporters can evolve novel substrates through generalist intermediates, either by increasing a weak activity or by establishing a new one. This coincides with what has been shown for soluble enzymes, and thus we show that membrane transporters can follow the same evolutionary logic (*Bergthorsson et al., 2007*; *Hughes, 1994*; *Aharoni et al., 2005*; *Des Marais and Rausher, 2008*; *Näsvall et al., 2012*). Looking at single adaptive mutations, we quantify the fitness trade-off between novel and original substrates, indicating that each mutation impacts the original substrates differently. Some of the identified adaptive mutations are almost neutral toward the original substrates, indicating that these mutations could become present in a large population, even before a new substrate is encountered by this population. This 'standing genetic variation' is a hallmark of contemporary theories of genetic innovation, and can explain how new functionalities arise while the protein is still under purifying selection for its original function (*Soskine and Tawfik, 2010*). Furthermore, such standing variation primes organisms for rapid adaptation (*Barrett and Schluter, 2008*). Along the same vein, we show that five wild-type amino acid

transporters have inherent side reactivities. In the light of evolution, this is a feature rather than a bug, as it can prime these transporters to enhance the side reactivities either by mutation or duplication, as observed for soluble enzymes (*Copley et al., 2023*; *Khersonsky and Tawfik, 2010*). Overall, we show how transporter evolution could help organisms to explore ecological niches where novel transport activities give a fitness benefit, while still retaining the ability to thrive in the old environment.

# Methods

## Key resources table

| Reagent type (species) or resource | Designation | Source or reference | Identifiers | Additional information |
|---|---|---|---|---|
| Strain, strain background (*Saccharomyces cerevisiae*) | Δ10AA (original name 22Δ10AAα) | Gift from Guillaume Pilot (*Besnard et al., 2016*) | *MATa gap1-1 put4-1 uga4-1 Δcan1::HisGΔlyp1-alp1::HisGΔhip1::HisGΔdip5::HisG Δgnp1Δagp1 ura3-1* | The original 22Δ10AAα strain was labeled as *MATα*, but was found to be *MATa* in our laboratory (data not shown). |
| Strain, strain background (*Saccharomyces cerevisiae*) | Δ10ΔUH | This paper derived from Δ10AA | *MATa gap1-1 put4-1 uga4-1 Δcan1::HisGΔlyp1-alp1::HisGΔhip1::HisGΔdip5::HisG Δgnp1Δagp1Δura3::loxP Δhis3::kanMX* | The original 22Δ10AAα strain was labeled as *MATα*, but was found to be *MATa* in our laboratory (data not shown). Strains can be requested from the corresponding authors. |
| Strain, strain background (*Saccharomyces cerevisiae*) | Δ10ΔUH pAR-Ec611 | This paper | Δ10ΔUH pAR-Ec611 (*HIS3*) | Strains can be requested from the corresponding authors. |
| Strain, strain background (*Saccharomyces cerevisiae*) | Δ10ΔUH evol-*AGP1* | This paper | Δ10ΔUH pAR-Ec611 (*HIS3*)+p1::*AGP1YPet* (*URA3*)+p2 | Strains can be requested from the corresponding authors. |
| Strain, strain background (*Saccharomyces cerevisiae*) | Δ10ΔUH evol-*PUT4* | This paper | Δ10ΔUH pAR-Ec611 (*HIS3*)+p1::*PUT4YPet* (*URA3*)+p2 | Strains can be requested from the corresponding authors. |
| Strain, strain background (*Saccharomyces cerevisiae*) | Δ10AA pADHXC3GH-*GOI*s | This paper | Strains containing the pADHXC3GH plasmids containing various genes of interest, see plasmids | Strains can be requested from the corresponding authors. |
| Strain, strain background (*Saccharomyces cerevisiae*) | GA-Y319 | Gift from Chang Liu | MATa *can1 his3 Δleu2Δura3Δtrp1 flo1*+p1+p2 | |
| Recombinant DNA reagent | FDP-P10B2-A75-RZ-URA3 (plasmid) | Gift from Chang Liu (Addgene plasmid # 130874) | Plasmid containing the integration cassette for p1, the 10B2 artificial promoter, a hardcoded poly-A tail, and an *URA3* selection marker | |
| Recombinant DNA reagent | FDP-*AGP1YPet* (plasmid) | This paper | Integration cassette for p1; derived from FDP-P10B2-A75-RZ-URA3, where the *mKate* gene was replaced by *AGP1* with an N-terminal YPet fluorescent tag | Sequences are deposited in https://zenodo.org/records/10928101. Vectors can be requested from the corresponding authors. |
| Recombinant DNA reagent | FDP-*PUT4YPet* (plasmid) | This paper | Integration cassette for p1; derived from FDP-P10B2-A75-RZ-URA3, where the *mKate* gene was replaced by *PUT4* with an N-terminal YPet fluorescent tag | Sequences are deposited in https://zenodo.org/records/10928101. Vectors can be requested from the corresponding authors. |
| Recombinant DNA reagent | pAR-Ec611 (plasmid) | Gift from Chang Liu (Addgene plasmid # 130872) | Error-prone TP-DNA polymerase 1 | |
| Recombinant DNA reagent | pYEXC3GH (plasmid) | Gift from Raimund Dutzler and Eric Geertsma (Addgene plasmid # 49027) | *Saccharomyces cerevisiae* expression vector for FX cloning | |
| Recombinant DNA reagent | pADHXC3GH (plasmid) | This paper | Derived from pYEXC3GH, where the *GAL1* promoter was exchanged for the constitutive *Saccharomyces cerevisiae* ADH promoter. The *GOI* is expressed as a fusion protein with N-terminal 3C cleavage site, yeGFP, and His$_{10}$ tag | Sequences are deposited in https://zenodo.org/records/10928101. Vectors can be requested from the corresponding authors. |
| Recombinant DNA reagent | pADHXC3GH-*AGP1* (plasmid) | This paper | | Sequences are deposited in https://zenodo.org/records/10928101. Vectors can be requested from the corresponding authors. |

*Continued on next page*

*Continued*

| Reagent type (species) or resource | Designation | Source or reference | Identifiers | Additional information |
|---|---|---|---|---|
| Recombinant DNA reagent | pADHXC3GH-*BAP2* (plasmid) | This paper | | Sequences are deposited in https://zenodo.org/records/10928101. Vectors can be requested from the corresponding authors. |
| Recombinant DNA reagent | pADHXC3GH-*CAN1* (plasmid) | This paper | | Sequences are deposited in https://zenodo.org/records/10928101. Vectors can be requested from the corresponding authors. |
| Recombinant DNA reagent | pADHXC3GH-*HIP1* (plasmid) | This paper | | Sequences are deposited in https://zenodo.org/records/10928101. Vectors can be requested from the corresponding authors. |
| Recombinant DNA reagent | pADHXC3GH-*LYP1* (plasmid) | This paper | | Sequences are deposited in https://zenodo.org/records/10928101. Vectors can be requested from the corresponding authors. |
| Recombinant DNA reagent | pADHXC3GH-*MMP1* (plasmid) | This paper | | Sequences are deposited in https://zenodo.org/records/10928101. Vectors can be requested from the corresponding authors. |
| Recombinant DNA reagent | pADHXC3GH-*PUT4* (plasmid) | This paper | | Sequences are deposited in https://zenodo.org/records/10928101. Vectors can be requested from the corresponding authors. |
| Recombinant DNA reagent | pADHXC3GH-*AGP1*-N (plasmid) | This paper | *T1001A*=I334N | Sequences are deposited in https://zenodo.org/records/10928101. Vectors can be requested from the corresponding authors. |
| Recombinant DNA reagent | pADHXC3GH-*AGP1*-V (plasmid) | This paper | *A1234G*=I412V | Sequences are deposited in https://zenodo.org/records/10928101. Vectors can be requested from the corresponding authors. |
| Recombinant DNA reagent | pADHXC3GH-*AGP1*-NV (plasmid) | This paper | *T1001A, A1234G*=I334N, I412V | Sequences are deposited in https://zenodo.org/records/10928101. Vectors can be requested from the corresponding authors. |
| Recombinant DNA reagent | pADHXC3GH-*AGP1*-G (plasmid) | This paper | *C1451G*=A484G | Sequences are deposited in https://zenodo.org/records/10928101. Vectors can be requested from the corresponding authors. |
| Recombinant DNA reagent | pADHXC3GH-*AGP1*-T (plasmid) | This paper | *G1450A*=A484T | Sequences are deposited in https://zenodo.org/records/10928101. Vectors can be requested from the corresponding authors. |
| Recombinant DNA reagent | pADHXC3GH-*PUT4*-S (plasmid) | This paper | *T620C*=L207S | Sequences are deposited in https://zenodo.org/records/10928101. Vectors can be requested from the corresponding authors. |

## Media

*S. cerevisiae* strains were grown in YPD medium (Formedium; nonselective medium, autoclaved), YNB medium (6.9 g/L yeast nitrogen medium without amino acids [Formedium]+20 g/L glucose; selective medium, sterile filtered) or YB medium (1.9 g/L YNB without amino acids and without ammonium sulfate [Formedium]+20 g/L glucose; selective medium without nitrogen source, sterile filtered). To supplement specific nitrogen sources to YB media, stock solutions were prepared by filter sterilization.

For in vivo evolution, the following media were used: Cit evolution medium (YB supplemented with Ala, Asn, Asp, GABA, Gln, Glu, Gly, Ile, Leu, Met, Orn, Phe, Pro, Ser, Thr, Trp, Val at 60 µM each and Cit at 3.06 mM), Asp evolution medium (YB supplemented with Pro, GABA at 0.5 mM each and Asp at 3 mM), Glu evolution medium (YB supplemented with Pro, GABA at 0.5 mM each and Glu at 3 mM). The amino acids were purchased from Sigma-Aldrich.

## DNA plasmid construction

The plasmids used in this study are listed in Key resources table. Cloning was performed with standard molecular biology methods; pADHXC3GH and its derivatives were constructed via Gibson assembly (*Gibson et al., 2009*) and FX cloning (*Geertsma and Dutzler, 2011*). Site-directed mutagenesis (*Kunkel, 1985*) was also achieved by Gibson assembly, using gene-specific primers with base changes at the position of interest. The single-mutation genes were assembled from the plasmid backbone and two separate PCR fragments, while the double-mutation *AGP1*-NV variant was assembled from three PCR fragments. Plasmids were transformed and amplified in *E. coli* DB3.1. Individual clones were

picked, grown to saturation in antibiotic selective LB or TB liquid media, mini-prepped and sequence-confirmed by Sanger sequencing (*Sanger et al., 1977*) (Eurofins Genomics). Plasmid sequences are available at https://zenodo.org/records/10928101 (DOI10.5281/zenodo.8426062).

## Yeast strains and transformation

The yeast strains used in this study are listed in Key resources table. Yeast transformations were performed as described (*Gietz and Schiestl, 2007*). DNA extraction from yeast strains was performed by using glass beads. Briefly, cells were washed with 250 μL of sterile water, pelleted, and resuspended in 30 μL Zymolyase 20T (80 μg in total). Glass beads of 0.55 m diameter were added in 1:1 vol/wt ratio of Zymolyase to glass beads. The samples were incubated at 37°C for 30 min and vortexed for 1 min at maximum speed. Subsequently, they were incubated at 95°C for 5 min and cooled down on ice for 5 min. Finally, the cellular debris was pelleted and the supernatant could be used later on. Confirmation of the presence of the desired gene variant was performed by PCR amplification of the region of interest and subsequent Sanger sequencing (Eurofins Genomics).

Since the OrthoRep system requires two auxotrophic markers in the strain used for in vivo evolution (*Ravikumar et al., 2014*), the Δ10AA strain was modified by complete deletion of the *ura3-1* and *HIS3* open reading frames to yield the strain Δ10ΔUH. Using Δ10ΔUH, the in vivo evolution strains for *AGP1* and *PUT4* were constructed essentially as described in *García-García et al., 2020*. Briefly, *S. cerevisiae* Δ10ΔUH was transformed with pAR-Ec611 containing the error-prone TP-DNA polymerase 1. In parallel, the gene of interest was cloned into the FDP-P10B2-A75-RZ-URA3 plasmid containing the integration cassette for the linear cytoplasmic plasmid p1. This cassette was excised by digestion with *ScaI* and used for transformation of *S. cerevisiae* GA-Y319 (p1 manipulation strain). The recovered transformants were screened for recombinant p1 plasmids containing the gene of interest by PCR (*GOI*) and by agarose gel electrophoresis of complete DNA extracts. A positive colony was picked and used to transfer the recombinant p1 along with wild-type p2 into Δ10ΔUH pAR-Ec611 to yield the strains Δ10ΔUH evol-*AGP1* and evol-*PUT4*.

## In vivo evolution

### AGP1

A culture of Δ10ΔUH evol-*AGP1* was grown at 30°C for ca. 100 generations (14 passages, reinoculating 1:200) in 10 mL Cit evolution medium. At two time points, single colonies that were able to use Cit as the sole N-source were isolated by plating on YB+1 mM Cit agar dishes. To confirm the genotype-phenotype linkage, the evolved *AGP1* variants were cloned into the pADHXC3GH expression vector, sequenced, and introduced into naïve Δ10AA cells.

### PUT4

Parallel cultures of Δ10ΔUH evol-*PUT4* were grown at 30°C for ca. 33 generations (5 passages, reinoculating 1:100) in 2 mL Asp or Glu evolution medium or control medium without an additional nitrogen source. Thereafter, the growth on media containing Asp or Glu only (YB+Asp or Glu at 3 mM, no other nitrogen sources) was compared to the control cultures. These cultures were used to clone the bulk of *PUT4* variants into the pADHXC3GH expression vector and introduced into naïve Δ10AA cells. Transformants were plated on YB+Asp 1 mM or YB+Glu 1 mM agar dishes, and colonies showing growth on these selective media were used for sequencing.

Regarding the estimation of evolutionary space covered, the mutation rate is assumed to be approximately $10^{-5}$ per generation and nucleotide (*Ravikumar et al., 2014*; *Ravikumar et al., 2018*). The expected number of mutations in our experiments was about 5000 mutational events per passage for *AGP1* (10 mL culture volume and 1:200 dilution), and about 1000 mutational events per passage for *PUT4* (2 mL culture volume and 1:100 dilution). At a gene length of about 2000 bp, we expected to cover most single mutations already in the first or second passage (in the absence of selection). This is reflected in the finding that the strongly beneficial mutation L207S in *PUT4* was recovered in every selection on Asp or Glu tested.

## Growth assay

Single colonies of *S. cerevisiae* Δ10AA pADHXC3GH-*GOI* were inoculated in YB media supplemented with 4 mM $NH_4^+$ and 0.1 mg/mL ampicillin, and grown until late logarithmic phase. The cultures were

pelleted at 750×*g* for 10 min at 30°C and washed with YB media. The wells in the microplate were filled with the amino acids of interest to a final concentration of 2 mM and with culture cells to a final $OD_{600}$ of 0.04, to a final total well volume of 200 µL. Sterile water was added in the space between the wells to avoid evaporation. The prepared microplates included three biological replicates of the strains with the plasmid containing the *GOI* and one biological replicate of the strain with the empty vector. The absorbance in each well was measured at 600 nm in 30 min intervals without shaking of the microplate, at 30°C for 72 hr in a SpectraMax ABS Plus plate reader. Data are shown in this report as the specific growth rate for each biological replicate and the error bars describe the SEM. The growth rates were derived based on the Baranyi growth model (*Baranyi and Roberts, 1995*), using the *growthrates* package in *R*. The Huang model (*Huang, 2008*; *Huang, 2011*) was also tested, but deemed less suitable because of the higher number of interdependent parameters, which consumes part of the fitness effects. The code and raw data are available at https://zenodo.org/records/10928101 (DOI 10.5281/zenodo.8426062).

## In vivo transport assay

Single colonies of *S. cerevisiae* Δ10AA/pADHXC3GH-*GOI* were inoculated in YB media supplemented with 4 mM $NH_4^+$, and grown until stationary phase. Amount of the grown cultures was used to inoculate YB media supplemented with 3 mM urea so that $OD_{600}$ is 0.05, and grown until mid-logarithmic phase. The cultures were pelleted at 750×*g* for 10 min at 4°C and washed with either 100 mM KPC buffer (10 mM glucose, 100 mM $K_2HPO_4$, citric acid buffer, pH 5) in the case of subsequent uptake assays with Phe, Glu, and Cit, or 100 mM KP buffer (10 mM glucose, 13.4 mM $K_2HPO_4$, 86.8 mM $KH_2PO_4$, pH 6) in the case of subsequent uptake assays with Ala, GABA, and Gly. The washed cells were diluted in the respective ice-cold buffer at a final $OD_{600}$ of 0.56–1.05.

Prior to the assay, the cells were pre-warmed in a 30°C water bath for 5–10 min, and the uptake reactions were started by adding radiolabeled amino acid (PerkinElmer). For the uptake assays by *PUT4* variants, 1.1 µCi/mL of $^{14}C$-Glu at a final concentration of 1 mM, or 0.2 µCi/mL of one $^{14}C$-labeled amino acid (Ala, GABA, Gly) at a final concentration of 10 µM was added. For the uptake assays by *AGP1* variants, 0.9 µCi/mL of $^{14}C$-Cit at a final concentration of 1 mM, or 0.5 µCi/mL of one $^{14}C$-labeled amino acid (Phe, Glu) at a final concentration of 0.1 mM or 2 mM was added. The uptake of 2 mM Glu by the *AGP1*-G variant was performed with the addition of 4.5 µCi/mL of $^3H$-Glu at a final concentration of 2 mM. The cells were mixed by magnetic stirring and, at given time intervals, 100 µL samples were collected and rapidly filtered on a 0.45 µM pore size nitrocellulose filter, which was subsequently washed with a total of 4 mL ice-cold buffer. The filters were dissolved in 2 mL scintillation solution and vortexed before determining the radioactivity by liquid scintillation counting on a Tri-Carb 2800TR liquid scintillation analyzer (PerkinElmer).

For the calculation of the uptake rates in amol/(cell×min), the determination of the cell number was based on the translation of the $OD_{600}$ measurements to number of cells by counting on a Thoma counting chamber. The null hypothesis that the mean of the uptake rate of the *AGP1* wild-type variant is not different from that of the mutants was tested by using the one-way ANOVA (*Girden, 1992*), and in the cases of a p<0.05, the Dunnett's significance test (*Dunnett, 1955*) was performed (significance shown as asterisks). For the *PUT4* analysis, the null hypothesis was tested by using Student's t-test (*Student, 1908*) (p<0.05).

## Localization assay

Living cells were washed in Isotope Buffer as described above and placed on a slide. The fluorescence cell imaging was performed on a Zeiss LSM 710 confocal laser scanning microscope, equipped with a C-Apochromat 40×/1.2 NA objective with a blue argon ion laser (488 nm). Images were obtained with the focal plane positioned at the mid-section of the cells, showing continuity on the fluorescence signal at the periphery of the cell.

For the estimation of the surface expression of the transporters, 30–50 cells per variant were manually selected using the Fiji ellipse tool (*Schindelin et al., 2012*). The code for estimating the mean pixel intensity of the outline of each selected cell per µm² is available at https://zenodo.org/records/10928101 (DOI 10.5281/zenodo.8426062).

## Acknowledgements

The authors are grateful to Chang Liu for help and materials for the OrthoRep evolution platform. We also thank Michiel C Punter for help in analyzing fluorescence micrographs, as well as Arjen M Krikken and Ida van der Klei for sharing laboratory resources and protocols.

## Additional information

### Funding

| Funder | Grant reference number | Author |
|---|---|---|
| Horizon 2020 Framework Programme | 847675 | Sebastian Obermaier |

The funders had no role in study design, data collection and interpretation, or the decision to submit the work for publication.

### Author contributions

Foteini Karapanagioti, Conceptualization, Data curation, Formal analysis, Validation, Investigation, Visualization, Methodology, Writing – original draft, Project administration, Writing – review and editing; Úlfur Águst Atlason, Investigation, Methodology; Dirk J Slotboom, Supervision, Writing – original draft, Writing – review and editing; Bert Poolman, Resources, Supervision, Funding acquisition, Writing – original draft, Project administration, Writing – review and editing; Sebastian Obermaier, Conceptualization, Data curation, Supervision, Validation, Investigation, Visualization, Methodology, Writing – original draft, Project administration, Writing – review and editing

### Author ORCIDs

Foteini Karapanagioti ⓘ http://orcid.org/0009-0007-1700-4867
Dirk J Slotboom ⓘ https://orcid.org/0000-0002-5804-9689
Bert Poolman ⓘ http://orcid.org/0000-0002-1455-531X
Sebastian Obermaier ⓘ http://orcid.org/0000-0002-1238-4502

Reviewer #1 (Public review): https://doi.org/10.7554/eLife.93971.3.sa1
Reviewer #2 (Public review): https://doi.org/10.7554/eLife.93971.3.sa2
Reviewer #3 (Public Review): https://doi.org/10.7554/eLife.93971.3.sa3
Author response https://doi.org/10.7554/eLife.93971.3.sa4

## Additional files

### Supplementary files
• MDAR checklist

### Data availability
Code and raw data are available on Zenodo (https://zenodo.org/records/10928101).

The following dataset was generated:

| Author(s) | Year | Dataset title | Dataset URL | Database and Identifier |
|---|---|---|---|---|
| Karapanagioti F, Atlason UA, Slotboom DJ, Poolman B, Obermaier S | 2024 | Fitness landscape of substrate-adaptive mutations in evolved APC transporters | https://doi.org/10.5281/zenodo.8426062 | Zenodo, 10.5281/zenodo.8426062 |

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
