## [Editor Report · eLife assessment]

This **important** manuscript describes experimental evolution experiments using a novel genetic system in yeast, showing that solute carrier transporters can incorporate additional functionality through the introduction of point mutations to either the ligand binding site or gating helices. These findings provide **convincing** evidence to establish that for Amino Acid transporters of the APC-type family, evolution to recognize new substrates passes through generalist intermediates that can transport most amino acids.

---

## [Referee Report · Reviewer #1 (Public review)]

Summary:

The evolution of transporter specificity is currently unclear. Did solute carrier systems evolve independently in response to a cellular need to transport a specific metabolite in combination with a specific ion or counter metabolite, or did they evolve specificity from an ancestral protein that could transport and counter transport most metabolites. The present study addresses this question by applying selective pressure to *Saccharomyces cerevisiae* and studying the mutational landscape of two well characterised amino acid transporters. The data suggest that AA transporters likely evolved from an ancestral transporter and then specific sub families evolved specificity depending on specific evolutionary pressure.

Strengths:

The work is based on sound logic and the experimental methodology is well thought through. The data appear accurate, and where ambiguity is observed (as in the case of citruline uptake by AGP1), in vitro transport assays are carried out. to verify transport function.

Weaknesses:

The revisions have substantially strengthened the conclusions based on the results of this study. Follow up studies will no doubt try to rationalise/identify if specific mutational hot-spots exist within the APC fold that explain the specialisation observed in mammals (neurotransmitter vs. metabolic) for example.

---

## [Referee Report · Reviewer #2 (Public review)]

Summary:

This paper describes evolution experiments performed on yeast amino acid transporters aiming at the enlargement of the substrate range of these proteins. Yeast cells lacking 10 endogenous amino acid transporters and thus being strongly impaired to feed on amino acids were again complemented with amino acid transporters from yeast and grown on media with amino acids as the sole nitrogen source.

In the first set of experiments, complementation was done with seven different yeast amino acid transporters, followed by measuring growth rates. Despite most of them having been described before in other experimental contexts, the authors show that many of them have a broader substrate range than initially thought.

Moving to the evolution experiments, the authors used the OrthoRep system to perform random mutagenesis of the transporter gene while it is actively expressed in yeast. The evolution experiments were conducted such that the medium would allow for poor/slow growth of cells expressing the wt transporters, but much better/faster growth if the amino acid transporter would mutate to efficiently take up a poorly transported (as in case of citrulline and AGP1) or non-transported (as in case of Asp/Glu and PUT4) amino acid.

This way and using Sanger sequencing of plasmids isolated from faster-growing clones, the authors identified a number of mutations that were repeatedly present in biological replicates. When these mutations were re-introduced into the transporter using site-directed mutagenesis, faster growth on the said amino acids was confirmed. Growth phenotype were confirmed by uptake experiments using radioactive amino acids; corresponding correlation plots show that the assays based on growth rates versus radioactive uptake assays indeed can explain the effect of the mutations to a large extent.

When mapped to Alphafold prediction models on the transporters, the mutations mapped to the substrate permeation site, which suggests that the changes allow for more favorable molecular interactions with the newly transported amino acids.

Finally, the authors compared growth rates of the evolved transporter variants with those of the wt transporter and found that some variants exhibit a somewhat diminished capacity to transport its original range of amino acids, while other variants were as fit as the wt transporter in terms of uptake of its original range of amino acids.

Based on these findings, the author conclude that transporters can evolve novel substrates through generalist intermediates, either by increasing a weak activity or by establishing a new one.

Strengths:

The study provides evidence in favour of an evolutionary model, wherein a transporter can "learn" to translocate novel substrates without "forgetting" what it used to transport before. This evolutionary concept has been proposed for enzymes before, and this study shows that it also can apply to transporters. The concept behind the study is easy to understand, i.e. improving growth by uptake of more amino acids as nitrogen source. In addition, the study contains a large and extensive characterization of the transporter variants, including growth assays and radioactive uptake measurements. The authors performed experiments as part of the revision to show that the studied mutations do not greatly change surface expression of the transporters. Further they showed that in the absence of the evolutionary pressure, overexpression of the mutants versus the wildtype transporters does not affect growth rates, which is important to assess. Finally, the authors make careful conclusions saying that in real life, the evolutionary landscape is way more complex than under these "reductive" laboratory conditions with a strain lacking ten natively expressed amino acid transporters and being selected on a single amino acid in a defined medium.

Weaknesses:

The authors took a genetic gain-of-function approach based on random mutagenesis of the transporter. While this experimental approach is suited to find some gain-of-function variants for some of the amino acids, it has also its inherent limitations, the most important being that loss-of-function mutants are not sampled (though they might be interesting) and that mutagenesis is entirely random, thus not targeted. These weaknesses cannot be easily overcome other than by restarting the entire study and conducting for example deep mutational scanning experiments. The authors have done what they could do within the scope of this study to make this manuscript as complete and rigorous as possible.

---

## [Referee Report · Reviewer #3 (Public Review)]

The goal of the current manuscript is to investigate how changes in transporter substrate specificity emerge in response to a novel selective pressure. The authors investigate the APC family of amino acid transporters, a large family with many related transporters that together cover the spectrum of amino acid uptake in yeast.

The authors use a clever approach for their experimental evolutions. By deleting 10 amino acid uptake transporters in yeast, they develop a strain that relies on amino acid import by introduced APC transporters under nitrogen limiting conditions. They can thus evolve transporters towards transport of new substrates if no other nitrogen source is available. The main takeaway from the paper is that it is relatively easy for the spectrum of substrates in a particular transporter of this family to shift, as a number of single mutants are identified that modulate substrate specificity. In general, transporters evolved towards gain-of-function mutations (better or new activities) also confer transport promiscuity, expanding the range of amino acids transported.

The data in the paper support the conclusions, and the outcomes (evolution towards promiscuity) agree with the literature available for soluble enzymes. The authors do a good job in the discussion of relating the lessons of the current study to natural evolution.

---

## [Author Response]

The following is the authors’ response to the original reviews.

(1) The authors should show (i) whether the variants exhibit the same surface expression as wildtype and (ii) whether changes of surface expression (e.g. wt transporter expressed low and high) alters growth rates under conditions where growth depends on amino acid uptake. The authors say that the uptake of radioactive substrate and the overall fitness coincide (Figures 5 and 6), but it would be good to quantify the correlation, perhaps by using a scatterplot and linear regression.

We thank the reviewer for the questions and proposals. The comparison of the surface expression between the transporter-expressing variants was added to the manuscript (Figure 3- Figure supplement 1 and 2). In the case of the *AGP1* variants it was calculated that surface expression between the evolved mutants and the wild-type is similar, indicating that the transporter overexpression has no impact on the growth rate per se. The same analysis for the *PUT4* variants showed significant difference, with the *PUT4*-S variant seemingly expressed more than the wild-type. However, that does not seem to affect the uptake effect of the mutation in the cases of the original substrates of Ala, Gly and GABA, since in those cases the transporter activity for the evolved variant is substantially decreased (Figure 5). Thus, the variation on the surface expression between the mutant and the wild-type, which could be attributed to the small sample size and the inherent limitations of the analysis (imaging of a culture with cells in different planes), is not expected to interfere with the reported results.

Additionally, a scatterplot accompanied with a linear regression curve describing the connection between the overall fitness and uptake of 2 mM radioactive substrates was added to the manuscript, as advised (Figure 5- Figure supplement 2). In both cases of 2 mM Phe or Glu, the regression model explains 60-70% of the variation observed in the uptake rate of the amino acids by the different variants if changes in the uptake rate are dependent on changes in the fitness.

(2) The authors should further investigate to what extent the (over)expression of wildtype versus variant transporters impacts growth rates. I would recommend such experiments being done under conditions where nitrogen uptake does not depend on amino acid uptake. I could imagine that some of the fitness data are confounded by the general effects of mutations on growth rates. More concretely, I could imagine that overexpression of e.g. the AGP1-G variant is less of a burden for the yeast cells and would allow to grow them better in general. This could explain why its overall fitness is close to wt, whereas other variants exhibit diminished fitness (Fig. 4A).

The growth curves of all transporter variant cultures in the absence of selection for amino acid uptake have been presented in Figure 4 - Supplement figure 1. As proposed, the growth rates of the variants in medium with ammonium as nitrogen source were calculated and presented in Figure 3- Supplement figure 1 and 2. For both cases of *AGP1* and *PUT4* expressing variants, statistical analysis showed no significant difference between the mutants and the wild-type.

(3) It is quite remarkable that the PUT4-S variant has such a dramatically enlarged substrate spectrum. In addition, the fitness losses for Alanine and GABA are rather small. This striking finding asks the question of why yeast has not evolved this much better/more efficient variant in the first place?

We thank the reviewer for this very good question. We now included an explanation in the Discussion, but to give a short answer here: One should keep in mind that we used a 10-gene deletion strain to select for given mutants. Wild-type cells have a wide spectrum of substrates through the use of many amino acid transporters, and their regulation is intricately tuned to achieve optimum transport under any environmental circumstance. Broadening the spectrum of a single transporter thus would not lead to increased fitness. On the contrary, it would probably throw off this fine balance.

(4) It would be generally interesting which types of selections (transporter/amino acid combinations) were tried (maybe as part of the methods section). I could imagine that the examples that are shown in the paper are the "tip of the iceberg", and that many other trials may have failed either because the cultures died, or the identified clones would grow faster due to mutations outside of the plasmid. It would be helpful for researchers planning such experiments in the future to be made aware of potential stepping stones.

The issues raised here are spot-on, as we actually did test the evolution of *PUT4* towards transport of other amino acids than the two mentioned in the report. Aside from the successful Asp and Glu, we ran parallel cultures selecting for transport of Gln, Thr, Trp, Tyr, and Cit. Neither of these evolution regimes led to increased growth phenotypes that were linked to the evolved gene, and we did not investigate these cultures further. At this point, we cannot fully explain this result, which is why we decided to omit it from the report. The L207S variant of *PUT4* was later shown to indeed support growth on Gln, Thr, and Cit. Therefore, we speculate that the reason for not evolving this mutant in the respective evolution cultures was that the fitness gain in these amino acids was not large enough to be sufficiently enriched in the course of the evolution trial. Given that the Δ10AA strain still harbors nine amino acid transporter genes in its genome, it is conceivable that upregulation of some of these genes causes growth in some amino acids, prohibiting the selection of mutations in *PUT4* (e.g., by mutations outside the plasmid, as the reviewer aptly suggested). We deemed these (negative) results not appropriate for the manuscript, as our main focus was characterizing the fitness effects of single mutations, not the laboratory evolution process of obtaining the mutants.

(5) The authors took a genetic gain-of-function approach based on random mutagenesis of the transporter. In such approaches, it is difficult to know which mutation space is finally covered/tested, and information that can be gained from loss-of-function analyses is missed. Accordingly, the outcome is somewhat anecdotal. To provide an idea of the mutational landscape accessible, the authors could perform NGS of cultures without any selective pressure, and report the distribution of missense variants in the population.

We very much appreciate the interest in the details of the mutagenesis. Based on the information given in the original OrthoRep publications (e.g., Ravikumar et al., DOI: 10.1016/j.cell.2018.10.021; mutation rate approx. 10-5 per generation and nucleotide), we calculated the expected number of mutations per passage in our experiments. For *AGP1*, it is about 5000 mutational events per passage (10 mL culture volume and 1:200 dilution), and for *PUT4*, it is about 1000 mutational events per passage (2 mL culture volume and 1:100 dilution). At a gene length of about 2000 bp, we expect to cover most single mutations already in the first or second passage (in the absence of selection). This is reflected in the result that the strongly beneficial mutation L207S in *PUT4* was recovered in every selection on Asp or Glu we tested. We included this information in the Methods section.

That said, the present study was consciously designed to research gain-of-function mutations, as we wanted to know if and how membrane transporters can evolve new substrate specificities without losing the original functions. Our approach was chosen to reflect as close as possible a natural scenario where a microorganism encounters a new ecological niche (a new nutrient to be transported). At the same time, we included selective pressure to keep the capacity to thrive in the original niche (to assimilate an ancestral nutrient). This approach is designed to specifically *select against* any loss-of-function mutations, which is in line with most modern theories about evolution of protein function (excellently reviewed in Soskine and Tawfik, DOI: 10.1038/nrg2808). We find that this approach gives a good idea how transporters could evolve new functions in a natural setting. By engineering single mutations in the wild-type background of the transporters, we show the fitness effects of different single mutations - this finding thus does not depend on the mutational landscape that is covered in the experiment.

(6) The authors do not discuss the impact of these mutations on transport rates/kinetics, which are known to play a role in substrate selection in solute carriers (https://www.nature.com/articles/s41467-023-39711-y). Do the authors think ligand binding/recognition is more important than kinetic selection in the evolution of function?

Indeed, the observed phenotypes can stem from both changes in transport rate and changes in substrate binding. In our opinion, both are perfectly possible explanations for the behavior of evolved transporter variants. We are not discussing this in the manuscript as the weak transport of the novel substrates in the wild-type transporters did not allow us to unambiguously assign one or the other. Yet, we can lend minor circumstantial evidence pointing towards substrate affinity being the more important factor in evolving a new activity in transporters: Overall transport rate (for original substrates) declined in most evolved transporters. Therefore, it is a bit less likely that improved transport rate allowed novel substrates to be used as a nutrient. However, this is not to say that both processes can occur (even side by side).

(7) Ultimately, what are the selective pressures that drive transporter function? The authors pose this question but don't fully develop the idea. Would promiscuous variants still be selected for if the limiting nitrogen source was taken up by the cell via a different pathway (i.e. ammonium or perhaps arginine)?

Evolution and regulation of transporters is a very complex system, and we simplify this system in our single-transporter/single-amino acid approach. In nature, the selective forces are assumed to be much smaller than in our system, and multiple selective pressures might occur at the same time (maybe even in opposite directions). Therefore, such predictions are beyond the scope of the present study. To put it shortly, yeasts (and other organisms) have evolved the capacity to transport all natural amino acids. Yet, to actually allow fine-tuned regulation of transport of each individual amino acid, narrow- and broad-range transporters have evolved, including a lot of redundancy. This means that the question posed cannot be answered by yes or no, but by “it depends”.

(8) Amino acids are a special class of metabolites, in that they all have the same basic structure. Thus, transport systems really only need to recognize the amino and carboxyl groups with high fidelity, and can modulate the side chain binding site to increase specificity. This was demonstrated in a bacterial APC transporter (https://www.nature.com/articles/s41467-018-03066-6#Sec2). Is this why the APC fold is largely responsible for AA uptake in biology?

Indeed, typically, APC-type amino acid transporters bind the amino and carboxyl groups in the same position by backbone interactions. Therefore, this might be an ancestral feature of the APC superfamily and explain why this group represents the main group of amino acid transporters.

(9) There isn't much discussion on the location of the mutations with respect to binding site vs. gating helices. Are there hotspots of mutations within the APC, and areas where variation is poorly tolerated? It would be helpful to briefly review what is known about mutations that change amino acid specificity in the APC family. My impression is that other studies applying rational mutagenesis have also shown that single-site mutations in the binding pocket alter substrate specificity - are these analogous to the L207 in PUT4? PUT4: I64T comes up in 3 of 5 selections. Did the authors consider a closer analysis of this mutation, and if not, why?

We agree that it would be helpful to determine hotspots of mutations in APC transporters that lead to changes in selectivity. However, we feel that the current literature does not lend enough data to support an extended analysis of such hotspots. Conversely, the natural sequences of APC transporters are not similar enough to determine which residues are responsible for a certain selectivity profile. There are however some studies on site-directed mutagenesis, as mentioned by the reviewer. A short summary of those is discussed in the revised paper. Interpretation of the previous studies under the light of our results suggests that the evolutionary evolved sites derived in our work play a significant role in substrate selectivity and transporter function within the superfamily of the APC transporters.

As to the question why we did not include the I64T mutation in our experiments: this mutation lies within the poorly defined N-terminus of the protein, which is not part of the transmembrane core. We therefore deemed this residue as probably not connected to the specificity of the protein; it might be related to the protein’s stability in the cell, as the termini of transporters are known to be important for post-translational regulation, especially vacuolar degradation.

(10) What do we learn about the APC fold that informs our understanding of where substrate specificity arises in this fold? Do the authors think all SLC folds are equally capable of adaption, or are some more evolutionary-ready than others? An evolutionary analysis of these transporters to gain insights into whether the identified substitutions also occurred during natural evolution under real-life conditions would further strengthen the manuscript. Could the authors provide a sense of how similar the 18 yeast amino acid transporters are, such as sequence alignments or a matrix of pairwise sequence identity/similarity? Are they very diverged, or is the complement of amino acid substrates covered by a rather conserved suite of transporters?

We do not want to make bold statements about adaptive evolution in other SLC folds, but we consider it not unlikely that a similar approach will lead to similar conclusions in other transporters.

As advised, a pairwise identity matrix was added to the manuscript (Figure 1–figure supplement 2).

As to the proposed analysis focusing on natural occurrence of the mutations we found: we have indeed looked into this, but have not found evidence of such mutations. This is actually expected, as our selection regime puts “unnatural” selective pressures on a single transporter in isolation, which in reality co-evolved with a whole suite of other transporters that already have the capacity to transport all amino acids. Therefore, it is unlikely that the same mutations would happen in a natural setting. Our study is designed to capture evolution where a completely novel substrate is encountered, for which no transport mechanism has evolved yet.

(11) Throughout: some of the bar graphs show individual data points, but others do not (Figure 3, Figure 5). These should be shown for all experiments.

We thank the reviewer for the comment. In the revised version of the manuscript, we included individual data points in all bar graphs.

(12) For bar graphs in which no indication of significance is shown, does this mean that p>0.05? Comparisons that are not significant (p>0.05) should be indicated as such.

We thank the reviewer for the comment. In the revised version of the manuscript, we indicated in the legends that in cases of no significant difference (*p* > 0.05) between the wild-type and the evolved variants, no asterisks are shown.

(13) Figure 5, Figure 6: Are the three confocal images just three different fields of view? It might be useful to include a zoom-in on a single representative cell, as it is hard for the reader to see to evaluate the membrane localization.

In the revised version of the manuscript, we clarified that the three confocal images represent three different cultures, as each variant was tested in triplicates. We also included a zoom-in of a representative cell, as suggested.

(14) In the main text, page 9, the conditions used for each experimental evolution are not clear ("nitrogen limiting mixture of amino acids (1 mM final concentration))". I think this is an important detail, since the mixtures are quite different for the more promiscuous vs. the more selective transporter, and it would be helpful if this was described more clearly in the main text.

We thank the reviewer for the comment. We have included further clarification in the revised manuscript.

(15) Figure 1-Supplement 1 and Figure 4 Supplement 4 - can't read the figure labels. Try labeling columns and rows rather than individual plots.

We have taken the proposal into account and revised the proposed Figures accordingly.

(16) Page 9: "The transporter gene was sequenced and re-introduced into Delta-10AA cells." Was the plasmid isolated, sequenced, and re-introduced, or was the gene cut-and-pasted into a new vector backbone?

In the revised manuscript we have clarified that the gene was sequenced and then cloned into the expression vector and re-introduced into naïve Δ10AA cells.